# PtdInsP$_2$ and PtdSer cooperate to trap synaptotagmin-1 to the plasma membrane in the presence of calcium

Ángel Pérez-Lara[1†], Anusa Thapa[2†], Sarah B Nyenhuis[2], David A Nyenhuis[2], Partho Halder[1], Michael Tietzel[3], Kai Tittmann[3], David S Cafiso[2*], Reinhard Jahn[1*]

[1]Department of Neurobiology, Max Planck Institute for Biophysical Chemistry, Göttingen, Germany; [2]Department of Chemistry and Center for Membrane Biology, University of Virginia, Charlottesville, United States; [3]Department of Molecular Enzymology, Göttingen Center for Molecular Biosciences, Georg-August University Göttingen, Göttingen, Germany

**Abstract** The Ca$^{2+}$-sensor synaptotagmin-1 that triggers neuronal exocytosis binds to negatively charged membrane lipids (mainly phosphatidylserine (PtdSer) and phosphoinositides (PtdIns)) but the molecular details of this process are not fully understood. Using quantitative thermodynamic, kinetic and structural methods, we show that synaptotagmin-1 (from *Rattus norvegicus* and expressed in *Escherichia coli*) binds to PtdIns(4,5)P$_2$ via a polybasic lysine patch in the C2B domain, which may promote the priming or docking of synaptic vesicles. Ca$^{2+}$ neutralizes the negative charges of the Ca$^{2+}$-binding sites, resulting in the penetration of synaptotagmin-1 into the membrane, via binding of PtdSer, and an increase in the affinity of the polybasic lysine patch to phosphatidylinositol-4,5-bisphosphate (PtdIns(4,5)P$_2$). These Ca$^{2+}$-induced events decrease the dissociation rate of synaptotagmin-1 membrane binding while the association rate remains unchanged. We conclude that both membrane penetration and the increased residence time of synaptotagmin-1 at the plasma membrane are crucial for triggering exocytotic membrane fusion.

*For correspondence: cafiso@
virginia.edu (DSC); rjahn@gwdg.
de (RJ)

[†]These authors contributed equally to this work

## Introduction

Synaptic transmission in the nervous system is mediated by the regulated release of neurotransmitters from presynaptic nerve terminals. Before triggering, neurotransmitters are stored in synaptic vesicles. Upon arrival of an action potential, voltage-gated Ca$^{2+}$-channels open, leading to a rise in cytoplasmic Ca$^{2+}$ concentration, which triggers the fusion of synaptic vesicles with the presynaptic plasma membrane. The primary Ca$^{2+}$ sensor for synchronous neuronal exocytosis is synaptotagmin-1 (syt-1), a resident protein of synaptic vesicles that consists of a short luminal domain, a transmembrane α-helix and two cytoplasmic C2 domains, referred to as C2A and C2B. C2 domains were first identified in protein kinase C (PKC), and have since been detected in many diverse proteins (*Corbalan-Garcia and Gomez-Fernandez, 2014*). The C2 domains of syt-1 are composed of structurally conserved eight-stranded anti-parallel β-sandwiches of ~130 residues. These domains contain Ca$^{2+}$-binding loops at one end of the β-sandwich motif. Upon Ca$^{2+}$ binding, the electrostatic surface potential of the C2 domains is altered. As a result, Coulombic attraction develops towards the head groups of anionic phospholipids, leading to binding of the C2 domains to the membrane associated with completion of the Ca$^{2+}$ coordination sphere. Moreover, a conserved polybasic lysine patch located on the C2B domain also binds to anionic lipid in the absence of Ca$^{2+}$, being particularly attracted to multivalent phosphoinositides (PtdIns) (*Jahn and Fasshauer, 2012*;

**eLife digest** The human nervous system contains billions of neurons that communicate with each other across junctions called synapses. When a neuron is activated, the levels of calcium ions inside the cell rise. This causes molecules called neurotransmitters to be released from the neuron at a synapse to make contact with the second neuron. The neurotransmitters are stored inside cells within compartments known as synaptic vesicles and are released when these vesicles fuse with the membrane surrounding the cell.

Proteins called SNAREs regulate the membrane fusion process. These proteins assemble into bundles that help to drive vesicle and cell membranes together. Another protein called synaptotagmin-1 sticks out from the vesicle membrane and senses the levels of calcium ions in the cell to trigger membrane fusion at the right time. Synaptotagmin-1 has two regions that can bind to calcium ions, known as the C2 domains. When calcium ion levels rise, these domains insert into the cell membrane by binding to two fat molecules in the membrane called phosphatidylserine (PtdSer) and phosphatidylinositol 4,5-bisphosphate (PtdInsP$_2$). Synaptotagmin-1 also interacts with the SNARE proteins, but it is not known whether synaptotagmin-1 triggers fusion by binding directly to SNAREs, or by the way it inserts into the cell membrane.

Pérez-Lara et al. used several biophysical methods to investigate how synaptotagmin-1 binds to PtdSer and PtdInsP$_2$. The experiments show that these molecules bind to different regions of synaptotagmin-1 and work together to attach the protein to the cell membrane and insert the C2 domains. Calcium ions increase the affinity of synaptotagmin-1 binding to the cell membrane by making it harder for synaptotagmin-1 to separate from the membrane, rather than by increasing its ability to bind to it. Further experiments show that synaptotagmin-1 prefers to bind to membranes that contain PtdInsP2 over binding to the SNARE proteins.

Together, the findings of Pérez-Lara et al. suggest that calcium ions may trigger the release of neurotransmitters by trapping synaptotagmin-1 at the cell membrane rather than by directly affecting how it interacts with SNARE proteins. Further work will be needed to establish exactly how the SNARE proteins, PtdInsP2 and synaptotagmin-1 interact.

*Südhof, 2013*; *Chapman, 2008*; *Brose et al., 1992*; *Corbalan-Garcia and Gómez-Fernández, 2014*; *Südhof, 2012*).

Membrane binding of syt-1 has been widely studied (*Kuo et al., 2009*; *Wang et al., 2003*; *van den Bogaart et al., 2012*; *Vrljic et al., 2011*; *Kuo et al., 2011*; *Radhakrishnan et al., 2009*; *Li et al., 2006*; *Bai et al., 2004, 2002*; *Schiavo et al., 1996*), mostly using a cytoplasmic fragment including both C2 domains (termed the syt-1 C2AB fragment). However, the mechanism of syt-1 binding to the membrane remains a matter of controversy. For example, recent studies have reported that a double Lys-to-Ala mutation in the polybasic lysine patch (termed the KAKA mutant) does not alter the binding of the syt-1 C2AB fragment to vesicles containing phosphatidylserine (PtdSer) and/or phosphatidylinositol-4,5-bisphosphate (PtdIns(4,5)P$_2$) (*Radhakrishnan et al., 2009*). By contrast, the same mutant was reported in other studies to decrease the binding of the C2AB fragment to vesicles (*Vrljic et al., 2011*), even in the absence of PtdIns (*Li et al., 2006*), or to almost completely abolish the binding to soluble PtdIns(4,5)P$_2$, in either the absence or the presence of Ca$^{2+}$ (*van den Bogaart et al., 2012*). On the other hand, several studies suggest that not only the tandem C2AB fragment (*Vrljic et al., 2011*) but also the individual C2B (*van den Bogaart et al., 2012*) and C2A domains (*Guillen et al., 2013*; *Zhang et al., 1998*) might bind to PtdIns predominantly through the Ca$^{2+}$-binding loops in the presence of Ca$^{2+}$, and as a consequence, might compete with PtdSer to complete the coordination sphere of bound Ca$^{2+}$ (*Honigmann et al., 2013*).

To resolve these discrepancies, and to shed light on the binding mechanism of syt-1 to its main lipid effectors, PtdSer and PtdIns, we examined the kinetics of syt-1 binding to vesicles containing different amounts of PtdSer and PtdIns(4,5)P$_2$. We also used isothermal titration calorimetry (ITC) to measure the affinities of the C2AB fragment towards the head groups of PtdSer and PtdIns. Moreover, we performed nuclear magnetic resonance (NMR) experiments to determine the residues that are affected by binding to PtdSer and PtdIns(4,5)P$_2$, and electron paramagnetic resonance (EPR)

experiments to examine the orientation of the C2AB fragment with respect to the membrane surface. Our results demonstrate that PtdSer and PtdIns(4,5)P$_2$ act in a synergistic manner to enhance the penetration depth of syt-1 and to reduce the dissociation rate from the membrane. Furthermore, our data provide strong evidence that PtdIns binds to the polybasic lysine patch on the C2B domain and does not compete with PtdSer for sites in the Ca$^{2+}$-binding loops. Importantly, enhanced binding to the membrane in the presence of Ca$^{2+}$ is exclusively due to a decrease in the dissociation rate ($k_{off}$), whereas the association rate ($k_{on}$) remains unchanged. Although synergistic, the binding of the syt-1 C2AB fragment is dramatically different for PtdSer- and PtdIns(4,5)P$_2$-containing membranes, suggesting that the modes of binding to these two lipids are distinct in nature.

## Results

### PtdSer and PtdIns(4,5)P$_2$ play a synergistic role in membrane binding of synaptotagmin-1

In the first series of experiments, we carried out stopped-flow experiments in order to investigate the binding kinetics of the soluble domain of syt-1 (termed the C2AB fragment) to vesicles containing different molar ratios of PtdSer and PtdIns(4,5)P$_2$. We monitored Förster resonance energy transfer (FRET) between the tryptophan residues of the C2AB fragment and dansyl-labeled phospholipids that were incorporated into the liposomes. The time course of the fluorescence traces was fitted to a mono-exponential function in order to determine the observed rate constant ($k_{obs}$) *Figure 1a*. The measured $k_{obs}$ values were plotted as a function of vesicle concentration and fitted to a linear equation, in which the slope yields $k_{on}$, the y intercept yields $k_{off}$ and the $K_d$ was calculated by $k_{off}/k_{on}$ (*Hui et al., 2005*) (*Figure 1b*).

In our approach, we used a low protein concentration to avoid binding of the C2AB fragment to opposing membranes in the presence of Ca$^{2+}$ (resulting in vesicle clustering) (*Vennekate et al., 2012*) and other cooperative effects (*Figure 1c*). Our conditions allowed us to measure reliably the binding kinetics and affinity of the C2AB fragment to liposomes containing various molar ratios of PtdSer (0, 10 and 20%) and PtdIns(4,5)P$_2$ (0, 2.5 and 5%).

As expected, no binding was observable in the absence of PtdSer and PtdIns(4,5)P$_2$ and, unfortunately, the fluorescence traces for binding to vesicles containing 10% PtdSer or 2.5% PtdIns(4,5)P$_2$ could not be reliably fitted because of the low signal-to-noise ratio under our experimental conditions. However, vesicles containing both PtdSer and PtdIns(4,5)P$_2$ yielded a lower apparent dissociation constant ($K_d$) than vesicles containing either PtdSer or PtdIns(4,5)P$_2$ alone, even when the vesicles had similar charge density (*Figure 1d*). Surprisingly, no significant difference was observed in the bimolecular association rate constant ($k_{on}$) in almost all the tested lipid mixtures (*Figure 1e*), In other words, the decrease in affinity is solely due to a drop in the unimolecular dissociation rate constant ($k_{off}$) (*Figure 1f*). Thus, under these experimental conditions, the association rate for binding of the C2AB fragment to the vesicles is diffusion-limited and the binding equilibrium is controlled by the off-rate.

The results described above suggest that the C2AB fragment binds to PtdSer in a different manner than to PtdIns(4,5)P$_2$. To determine whether the orientation of the C2AB fragment differs when bound to membranes containing either PtdSer or PtdIns(4,5)P$_2$ in the presence of Ca$^{2+}$, we performed EPR experiments using spin-labeled variants of the protein. Insertion of the spin-labeled side chain R1 (see *Figure 2a*) into the hydrophobic core of the membrane alters the rotamer sampling and motion of the spin label. As consequence, the EPR spectra from membrane-embedded spin labels are significantly broadened when the C2AB fragment is fully bound to vesicles (*Figure 2a*). In addition, using progressive power saturation (*Freed et al., 2011*; *Frazier et al., 2003*), we determined the membrane insertion depth of labels attached to the Ca$^{2+}$-binding loops of the C2A domain (residues 173 and 234), the Ca$^{2+}$-binding loops of the C2B domain (residues 304 and 368) and the polybasic lysine patch (residue 329). When bound to bilayers containing 5% PtdIns(4,5)P$_2$, labels in the Ca$^{2+}$-binding loops exhibit more motional averaging than they do when bound to vesicles composed of either 20% PtdSer (*Figure 2a*) or 10% PtdSer + 2.5% PtdIns(4,5)P$_2$ (data not shown). This suggests a shallower membrane penetration for R1 when bound to bilayers composed of only PtdIns(4,5)P$_2$ rather than PtdIns(4,5)P$_2$ and PtdSer, in agreement with previous results of experiments based on continuous wave power saturation. As previously reported (*Herrick et al.,*

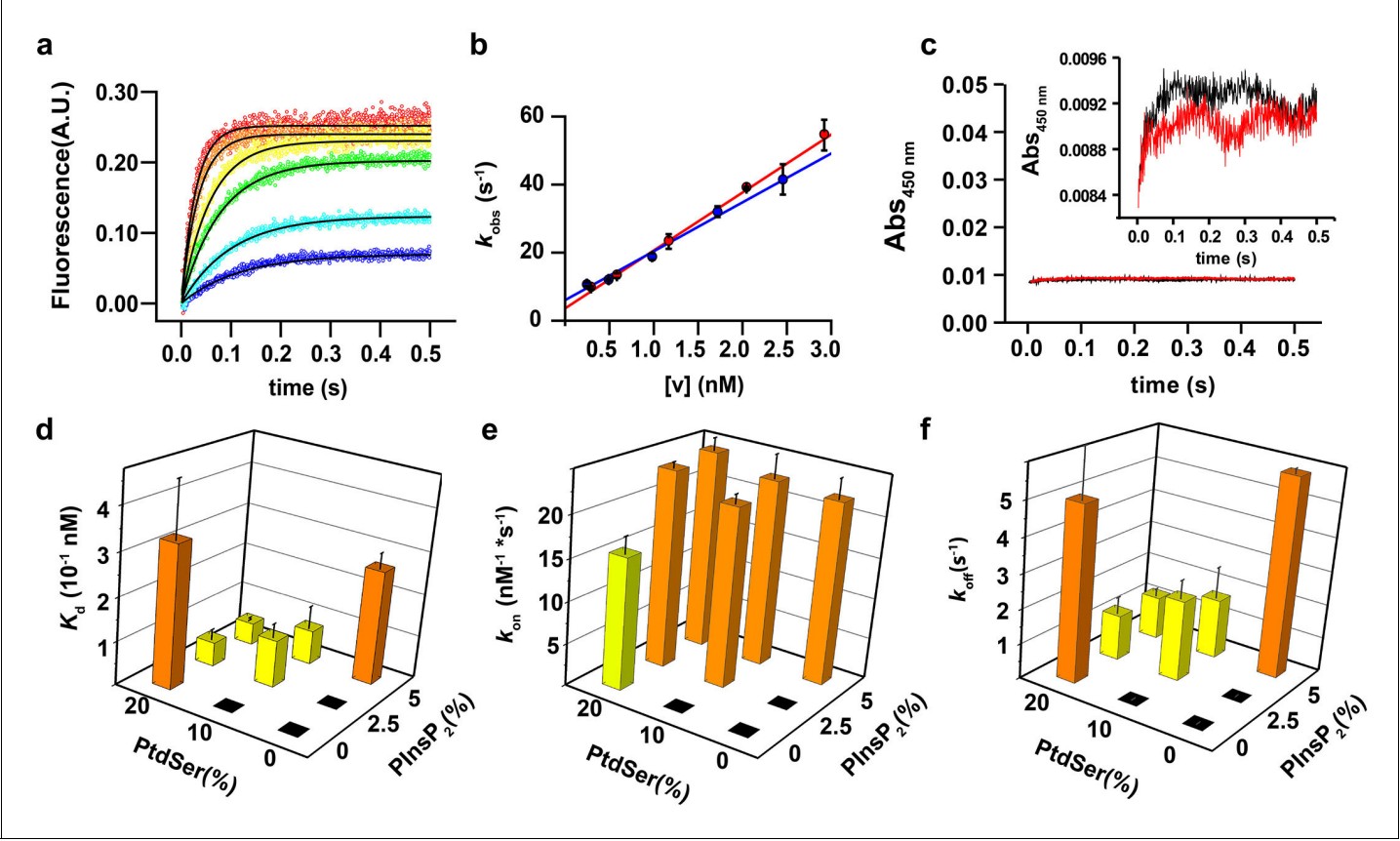

**Figure 1.** PtdIns(4,5)P$_2$ and PtdSer act cooperatively in the membrane binding of the C2AB fragment. (**a**) Representative time courses of dansyl emission for rapid mixing of 0.25 µM syt-1 C2AB fragment (final concentration) with increasing concentrations of large unilamellar vesicles (final liposome concentrations ranging between 0.3 nM and 3 nM) containing PtdChol:PtdSer (80:20, molar ratio), measured at 25°C. The apparent rate constant $k_{obs}$ was determined by fitting the traces to a monoexponential function (solid black lines). (**b**) The dependence of $k_{obs}$ on the vesicle concentration: data from two different sets of liposomes used for the stopped-flow experiment shown in (a). The non-zero y-intercept of the linear regression curve yields $k_{off}$ and the slope yields $k_{on}$. (**c**) Representative absorbance time course (n = 2) for rapid mixing of C2AB fragment (0.25 µM, final concentration) and vesicles (~2.5 nM). Absorbance was monitored at 450 nm for vesicles containing PtdChol/PtdSer/PtdIns(4,5)P$_2$ (75:20:5, molar ratio). No aggregation of vesicles was observed in our conditions, even when lipid mixtures with the highest affinity to C2AB fragment were used. (Inset) A scale-up of the traces showing that there is no vesicle aggregation in our conditions in concordance to the results of *Vennekate et al. (2012)*. (**d**) $K_d$ values, calculated as the ratio of $k_{off}/k_{on}$, (**e**) $k_{on}$ and (**f**) $k_{off}$ of C2AB fragment binding to liposomes containing different concentrations of PtdSer and PtdIns(4,5)P$_2$. All values were calculated from stopped-flow experiments carried out by rapid mixing of C2AB fragment with vesicles containing different amounts of PtdSer and/or PtdIns(4,5)P$_2$, and 100 µM free Ca$^{2+}$ at 25°C (n = 5–10). Higher affinities (lower $K_d$) were observed in the presence of both PtdSer and PtdIns(4,5)P$_2$, solely resulting from a decrease in $k_{off}$. Columns with the same color indicate values that were not significantly different from each other. Black boxes indicate either lack of binding or low binding that was too noisy for a reliable quantitative analysis.

2006; *Kuo et al., 2009*), R1 sidechains at sites 173, 234, 368 and 304 penetrate the membrane in the presence of PtdSer regardless of whether PtdIns(4,5)P$_2$ is present, but failed to deeply penetrate the membrane in vesicles containing only PtdIns(4,5)P$_2$. As shown in *Table 1*, the insertion depth of the Ca$^{2+}$-binding loops was 4–6 Ångstroms (Å) shallower in membranes composed of PtdIns(4,5)P$_2$ than in membranes containing PtdSer. By contrast, a label in the polybasic patch, 329R1, was closer to the membrane in PtdIns(4,5)P$_2$-containing vesicles than in PtdSer-containing vesicles, indicating that there is a difference in orientation. Using the power saturation data in *Table 1*, the orientations of C2B domain in the presence of membranes containing either PtdSer or PtdIns(4,5)P$_2$ were determined (see Materials and methods) and are shown in *Figure 2b*. The presence of PtdIns(4,5)P$_2$ is found to promote a tilt of the C2B domain, which is consistent with its interaction with the polybasic face. It should be kept in mind that the depth measurements are averaged values and that these models represent an average of what must be a dynamic system. Together with the stopped-flow

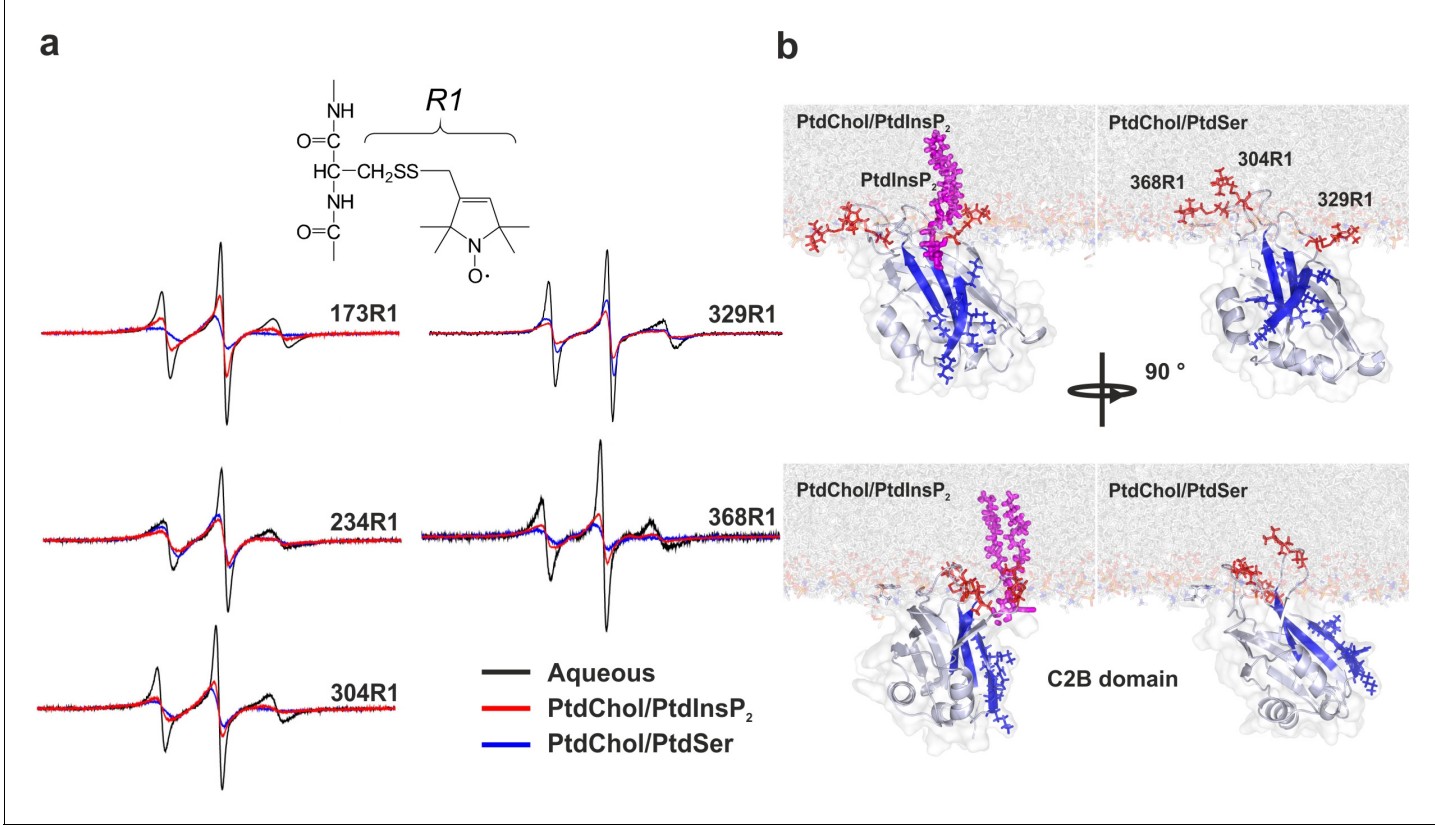

**Figure 2.** Orientation of the C2 domains bound to membranes containing either PtdSer or PtdIns(4,5)P$_2$. (a) EPR spectra from sites 173, 234, 304, 329 and 368 in solution or bound to bilayers containing PtdChol/PtdSer (80:20, molar ratio) or PtdChol/PtdInsP$_2$ (95:5, molar ratio). The broader spectra reflect diminished amplitudes and rates of R1 label motion when the C2AB fragment is bound to vesicles containing PtdChol/PtdIns(4,5)P$_2$ (95:5, molar ratio) or PtdChol/PtdSer (80:20, molar ratio) (n = 2–3). R1 is the spin-labeled side chain produced by derivatizing cysteine with the MTSL spin label. (b) Docking orientation of the C2B domain (PDB ID: 1K5W [*Fernandez et al., 2001*]) at the membrane interface, polybasic patch in blue. In this figure, the result from Xplor-NIH (Materials and methods) was aligned with a membrane simulation generated by CHARMM-GUI (*Jo et al., 2008*) for a bilayer with PtdChol:PtdInsP$_2$ or PtdChol:PtdSer at a molar ratio of 97:3 or 80:20, respectively. PtdIns(4,5)P$_2$ (pink sticks) was manually docked to the C2B domain. Blue sticks correspond to residues 321–326 and red sticks correspond to R1-labeled residues used in EPR experiments.

measurements described above, these data indicate that PtdSer and PtdIns(4,5)P$_2$ play a synergistic role in the binding of syt-1 to the membrane. When both PtdSer and PtdIns(4,5)P$_2$ are present, the Ca$^{2+}$-binding loops of syt-1 penetrate deeper into the bilayer than they do when the membrane contains PtdIns(4,5)P$_2$ alone, leading to a tighter binding of the C2AB fragment to the membrane, which is reflected by a decrease in the dissociation rate.

## PtdInsP$_2$ binding to the C2B domain polybasic patch

To further characterize the binding of the C2AB fragment to PtdIns(4,5)P$_2$ and PtdSer, we performed isothermal titration calorimetry (ITC) experiments in the presence of Ca$^{2+}$ using the head groups of PtdSer and PtdIns(4,5)P$_2$, O-phospho-L-serine (O-PSer) and inositol-1,4,5-triphosphate (InsP$_3$), respectively, as ligands for our titration instead of vesicles. Our rationale was that by using the head group of the phospholipid, we could determine the role of each specific phospholipid in the binding of the C2AB fragment to the membrane, avoiding the interferences of other phospholipids and unspecific electrostatic and hydrophobic interactions between the protein and the membrane. In our experiments, only small changes in the binding enthalpy were observed upon injection of O-PSer into a solution containing the C2AB fragment, precluding a quantitative assessment of the thermodynamics of binding. This did not change even when we used a short-chain analogue of PtdSer (1,2-hexanoyl-sn-glycero-3-phospho-L-serine, C$_6$-PtdSer) instead of O-PSer. By contrast, binding of InsP$_3$

**Table 1.** Depth parameters and approximate positions of spin labels attached to the C2AB fragment.

| Mutant | Lipid composition | Depth parameter (Φ) | Approx. distance to phosphate plane (Å)* |
|--------|-------------------|---------------------|-------------------------------------------|
| M173R1 | 20% PtdSer | +1.20 ± 0.10 | +8.8 |
| | 5% PtdIns(4,5)P$_2$ | −0.01 ± 0.20 | +5.5 |
| | 10% PtdSer + 2.5% PtdIns(4,5)P$_2$ | +0.66 ± 0.10 | +7.4 |
| F234R1 | 20% PtdSer | −0.10 ± 0.10 | +5.2 |
| | 5% PtdIns(4,5)P$_2$ | −1.50 ± 0.03 | −0.6 |
| | 10% PtdSer + 2.5% PtdIns(4,5)P$_2$ | +0.00 ± 0.10 | +5.5 |
| V304R1 | 20% PtdSer | −0.39 ± 0.20 | +4.3 |
| | 5% PtdIns(4,5)P$_2$ | −1.30 ± 0.10 | +0.6 |
| | 10% PtdSer + 2.5% PtdIns(4,5)P$_2$ | −0.23 ± 0.10 | +4.8 |
| T329R1 | 20% PtdSer | −2.00 ± 0.14 | −5.4 |
| | 5% PtdIns(4,5)P$_2$ | −0.80 ± 0.14 | +2.8 |
| | 10% PtdSer + 2.5% PtdIns(4,5)P$_2$ | −2.10 ± 0.12 | −7.3 |
| G368R1 | 20% PtdSer | +0.60 ± 0.10 | +7.2 |
| | 5% PtdIns(4,5)P$_2$ | −0.64 ± 0.05 | +3.6 |
| | 10% PtdSer + 2.5% PtdIns(4,5)P$_2$ | +0.18 ± 0.10 | +6.0 |

*Positive distances lie on the hydrocarbon side of a plane defined by the lipid phosphates; negative distances reside on the aqueous side. Distances were estimated using a calibration curve empirically determined as described previously (**Herrick et al., 2006**). Depth parameters are typically averages of two to four measurements and the error represents either standard deviations or errors propagated from the error in the measurement of ΔP½. These measurements were carried out at lipid concentrations high enough so that C2AB fragment is effectively completely membrane bound in the presence of 1 mM Ca$^{2+}$.

to the C2AB fragment was highly exothermic, showing a stoichiometry of 1:1 and an affinity of 14 ± 2 µM (**Figure 3a**).

To gain insight into the binding sites, we examined the average weighted chemical shift changes in a $^1$H-$^{15}$N HSQC NMR spectrum of $^{15}$N-labeled C2AB fragment upon addition of either InsP$_3$ or O-PSer in the presence of Ca$^{2+}$. In agreement with the lack of enthalpy changes, addition of O-PSer to the C2AB fragment did not induce significant changes in the chemical shifts (**Figure 3—figure supplement 1a**). By contrast, the addition of InsP$_3$ induced chemical shift changes primarily in the polybasic lysine patch of the C2B domain, with the highest chemical shifts observable for residues K325, K326 and K327, confirming previous reports which established the polybasic lysine patch of the C2B domain as the primary binding site of PtdIns(4,5)P$_2$ (**Vrljic et al., 2011**; **Li et al., 2006**; **Bai et al., 2004**; **van den Bogaart et al., 2012**) (**Figure 3b,c** and **Figure 3—figure supplement 1b**). In addition to the polybasic lysine patch, residues located in the Ca$^{2+}$-binding sites of the C2A and C2B domains exhibited significant but smaller chemical shift changes (**Figure 3b,c**).

To determine whether the Ca$^{2+}$-binding sites contribute to the overall binding affinity of InsP$_3$, we measured InsP$_3$ binding by ITC to C2AB-fragment mutants in which Ca$^{2+}$-binding was inactivated either in the C2A domain (C2aB: D178A, D230A, D232A) or in the C2B domain (C2Ab: D309A, D363A, D365A) or both (C2ab: D178A, D230A, D232A, D309A, D363A, D365A) (**Radhakrishnan et al., 2009**). The experiments were carried out in the presence and absence of Ca$^{2+}$ (**Figure 4a,b**), with the wild-type C2AB fragment as control. In the absence of Ca$^{2+}$, the binding affinity of the wild-type protein was significantly lower than in the presence of Ca$^{2+}$. This reduction is mediated by the Ca$^{2+}$-binding site of the C2B domain: it was abolished in the C2Ab and C2ab mutants but not in the C2aB mutant (**Figure 4d**). Apparently, the negatively charged side chains in the Ca$^{2+}$-binding site reduce the affinity of the negatively charged InsP$_3$ through unfavorable Coulombic interactions. Neutralizing these charges by Ca$^{2+}$-binding or, alternatively, by their deletion by mutagenesis are equally effective in preventing the thermodynamic penalty resulting from repulsive interactions. In line with that proposition, the lower affinities were mainly due to a decrease in binding enthalpy rather than to reduced entropy (**Figure 4—figure supplement 1**;

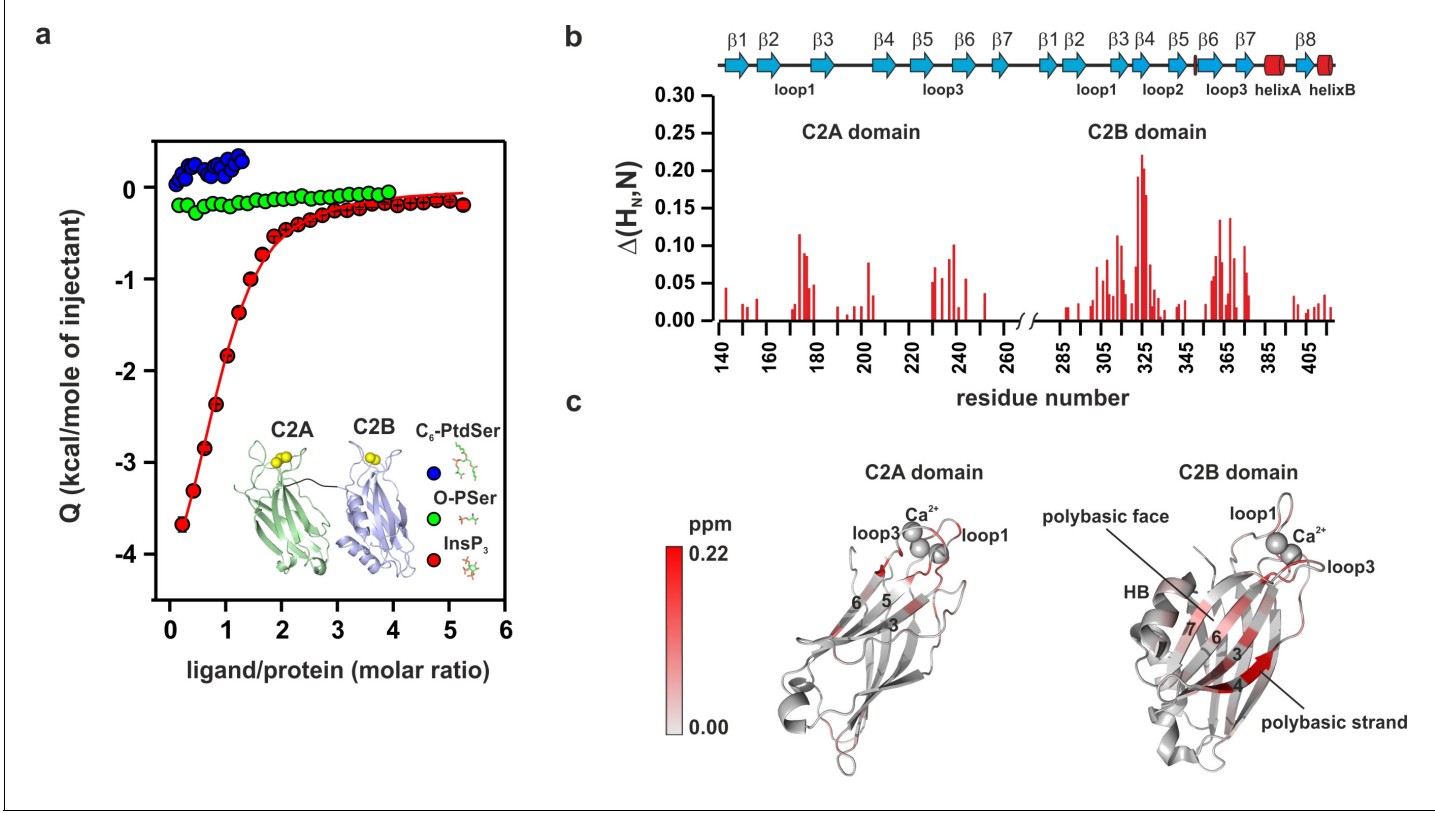

**Figure 3.** PtdIns(4,5)P$_2$ binds preferentially to the polybasic patch of the C2B domain. (**a**) Binding of the C2AB fragment to phospholipid head groups measured by isothermal titration calorimetry (ITC). Titration of ~50 µM C2AB fragment (50 mM HEPES, pH 7.4, 150 mM NaCl and 1 mM CaCl$_2$) with InsP$_3$ (inositol-1,4,5-triphosphate) (n = 3), O-phosphoserine (O-PSer) (n = 2) and 1,2-hexanoyl-sn-glycero-3-phospho-L-serine (C$_6$-PtdSer) (n = 2), in the presence of saturating Ca$^{2+}$ at 25°C. Only small heats were observed upon addition of C$_6$-PtdSer and O-PSer, while InsP$_3$ binds specifically ($K_d$ = 14 ± 2 µM) with a stoichiometry of 1:1. (**b**) Averaged-weighted chemical shifts ($\triangle H_N$, N) in $^{15}$N-$^1$H correlated HSQC NMR spectra of syt-1 in the presence of InsP$_3$ and Ca$^{2+}$. Chemical shift changes are widely distributed in the polybasic region of C2B domain. Small chemical shift are also seen in the calcium-binding loops of the C2B and C2A domains. Measurements were made under normal ionic strength (150 mM NaCl, 50 mM MES, pH = 6.3, and 3 mM Ca$^{2+}$) at a frequency of 600 MHz for protons. (**c**) Chemical shifts are color coded and mapped onto the structures of C2B and C2A domains according to the color bar (n = 2–3).

The following figure supplement is available for figure 3:

**Figure supplement 1.** Chemical shift of $^1$H-$^{15}$N HSQC of C2B domain in the presence of InsP$_3$ or O-PSer.

*Figure 4—figure supplement 2*), as one would expect for a primarily electrostatic effect. Nevertheless, no change of the binding stoichiometry was observed in any of the different conditions and mutants tested (*Figure 4c*). Next, we repeated the ITC titrations using the isolated C2A and C2B domains of syt-1. As expected, the C2B domain bound InsP$_3$ in a manner similar to the C2AB fragment. By contrast, binding of InsP$_3$ to the C2A domain yielded only small enthalpy changes, confirming that C2B is indeed the domain responsible for InsP$_3$ binding (*Figure 4e*).

Recently, it has been reported that, in the absence of Ca$^{2+}$, syt-1 binds preferentially to the SNARE complex rather than to membranes containing PtdIns(4,5)P$_2$ (*Brewer et al., 2015*). To test this hypothesis, we measured InsP$_3$ binding to the isolated C2B domain in the presence of SNARE complex and in the absence of Ca$^{2+}$ (to avoid precipitation of protein [*Dai et al., 2007*]) by ITC. The presence or absence of SNARE-complex made no difference to InsP$_3$ binding to the isolated C2B domain (*Figure 4f*), in agreement with previous observations from our group (*Park et al., 2015*). This evidence suggests that, under physiological conditions, the C2AB fragment binds preferentially to PtdIns(4,5)P$_2$. A recent study has reported three distinct syt–SNARE interfaces involving both C2A and C2B domains (*Zhou et al., 2015*). Therefore, we next repeated the experiment

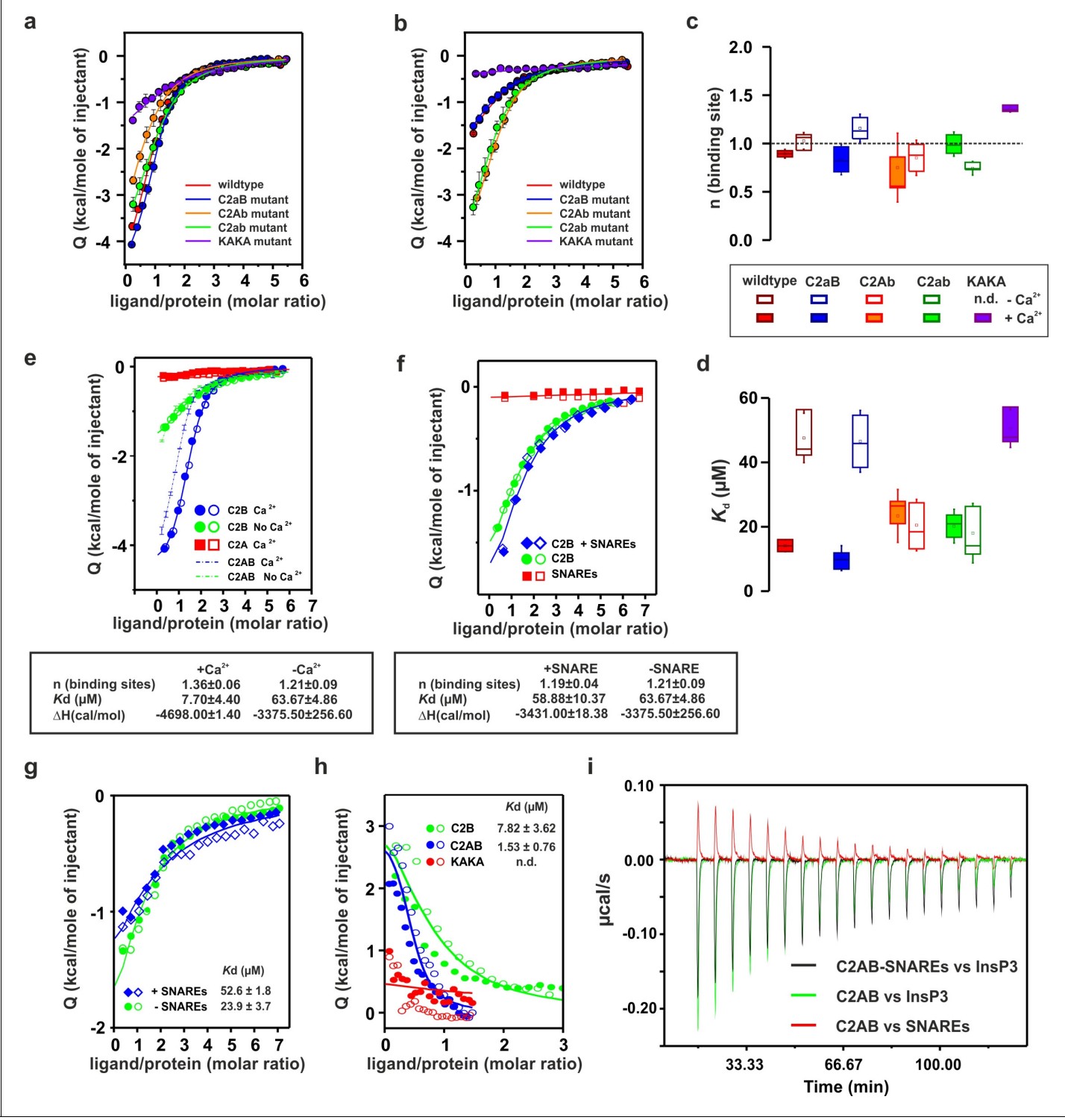

**Figure 4.** $Ca^{2+}$ increases the affinity of $PtdIns(4,5)P_2$ to the polybasic patch by means of shielding the negative charges of the $Ca^{2+}$-binding site in the C2B domain. (**a,b**) Representative ITC titrations (n = 3–6) of ~50 µM of the wild-type C2AB fragment, mutant proteins in which either one or both $Ca^{2+}$ binding sites are mutated, and the KAKA mutant in which some of the charges in the polybasic patch are removed. Titrations were carried out with $InsP_3$ as shown in *Figure 3a* in (**a**) the presence or (**b**) the absence of $Ca^{2+}$. Lines represent the fitting of the different titrations. (**c**) Number of binding sites determined from the fits of the ITC experiments of the different C2 domains shown in (**a**) and (**b**). All C2 domains show one binding site for $InsP_3$ in the presence or absence of $Ca^{2+}$. In absence of $Ca^{2+}$, binding of $InsP_3$ to the KAKA mutant was completely abolished. (**d**) Dissociation constants ($K_d$) calculated from the experiments shown in (**a**) and (**b**). In the presence of $Ca^{2+}$, only the KAKA mutant shows a significant decrease in the binding to

*Figure 4 continued on next page*

*Figure 4 continued*

InsP$_3$. In the absence of Ca$^{2+}$, mutations in the C2B Ca$^{2+}$-binding site rescue the binding to InsP$_3$, where as the KAKA mutant shows weak unspecific binding. (**e**) Titration of ~50 μM C2A domain (in the presence of Ca$^{2+}$) and C2B domain (in the presence or absence of Ca$^{2+}$) with InsP$_3$ (n = 2) compared to that of the C2AB fragment in same conditions (see legend). The C2B domain is responsible for InsP$_3$ binding to syt-1. (**f**) Titration of ~50 μM C2B domain–SNARE-complex (1:1, molar ratio) with InsP$_3$ in the absence of Ca$^{2+}$ (n = 2). The C2B domain binds preferentially to InsP$_3$ in physiological conditions. (**g**) Titration of ~15 μM C2AB domain/SNARE-complex (1:1, molar ratio) with InsP$_3$ in the absence of Ca$^{2+}$ (n = 2). We set the number sites to one for the fitting due to the uncertainty resulting from the low concentrations used. (**h**) Titration of SNARE-complex ~15 μM with different syt-1 fragments in the absence of Ca$^{2+}$ (n = 2). The $K_d$ values from (**g**) and (**h**) should be interpreted with caution because the low concentration used for the titrations resulted in a high uncertainty on the fitting. (**i**) Representative raw data for titration from (**g**) and (**h**). Interestingly, titration of the C2AB–SNARE complex presented an endothermic profile, whereas titration of InsP$_3$ presented an exothermic profile.

The following figure supplements are available for figure 4:

**Figure supplement 1.** Thermodinamic binding parameters of the different C2AB fragments.

**Figure supplement 2.** Representative titrations (n ≥ 3) of ~50 μM of the C2AB fragment used in *Figure 4a,b* in the presence or absence of Ca$^{2+}$.

described above using the C2AB fragment and lower protein concentration, ~15 μM instead of ~50 μM, to reduce the chances of protein aggregation. Interestingly, the presence of the SNARE complex resulted in a moderate decrease in affinity of InsP$_3$ to the C2AB fragment, in contrast to above results using the C2B domain (*Figure 4g,i*). To clarify this discrepancy, we decided to examine the syt–SNARE interaction using ITC. Our results show that the SNARE complex binds the C2AB fragment with higher affinity than it does the C2B domain, and that K326 and K327 play an important role in this binding (*Figure 4h,i*), in agreement with recent findings (*Brewer et al., 2015*; *Wang et al., 2016*).

Previous studies have shown that the mutation of these two Lys residues in the polybasic lysine patch to Ala (K326A and K327A, KAKA mutant) abolishes syt-1 binding to PtdIns(4,5)P$_2$ (*van den Bogaart et al., 2012*). Using ITC, we studied the binding of InsP$_3$ to the KAKA mutant in the presence and absence of Ca$^{2+}$. In the absence of Ca$^{2+}$, binding to InsP$_3$ was abolished (*Figure 4b*), as previously reported (*van den Bogaart et al., 2012*). However, in the presence of Ca$^{2+}$, the KAKA mutant was able to bind to InsP$_3$, but with lower affinity than the wild-type C2AB fragment (*Figure 4a,d*). To further investigate the binding of the KAKA mutant to the membrane in the absence of Ca$^{2+}$, we performed stopped-flow experiments in the absence of Ca$^{2+}$. No binding was detectable using FRET, probably because of the low concentration of protein used in our pseudo-first-order conditions, which did not provide enough signal for detection (data not shown). To overcome this limitation, we carried out a vesicle sedimentation assay to measure the fraction of protein bound to sucrose-loaded vesicles under true equilibrium conditions. Using sedimentation, the KAKA mutant was found to bind with reduced affinity to either PtdSer- or PtdIns(4,5)P$_2$-containing bilayers in the presence of Ca$^{2+}$ (*Figure 5a,b,c* and *Table 2*). Remarkably, the KAKA mutation produced only a small effect on the binding to PtdSer-containing bilayers (*Figure 5a*), but dramatically reduced the membrane affinity of C2AB fragment to 2% PtdIns(4,5)P$_2$ bilayers (*Figure 5b*). In these membranes, the KAKA mutant altered the membrane-binding free energy of C2AB fragment by 7.5 kJ/mole. In the absence of Ca$^{2+}$, the KAKA mutant failed to bind to PtdSer- or PtdIns(4,5)P$_2$-containing bilayers (*Figure 5d,e*). However, this variant did bind to membranes composed of both PtdSer and PtdIns(4,5)P$_2$ (*Figure 5f*), indicating a synergistic activity of PtdSer and PtdIns(4,5)P$_2$ towards syt-1 C2AB fragment binding.

The affinity between the C2AB fragment and membranes containing both PtdSer and PtdIns(4,5)P$_2$ (*Table 3*) in the absence of Ca$^{2+}$ indicates that at an accessible lipid concentration of 20 μM, 50% of the protein is bound. However, in full-length syt-1, the C2 domains are tethered near the membrane interface and thus experience a very high local lipid concentration. Assuming the domains lie within 4 nm of the interface, the effective membrane concentration seen by the C2 domains will be well over four orders of magnitude higher than 20 μM. This represents a shift in the binding free energy to favor membrane association by about 25 kJ/mole, may be even higher when considering that syntaxin and PtdIns(4,5)P$_2$ form clusters that increase the local concentration of PtdIns(4,5)P$_2$ in the plasma membrane (*Honigmann et al., 2013*; *van den Bogaart et al., 2011*). Of course, the local

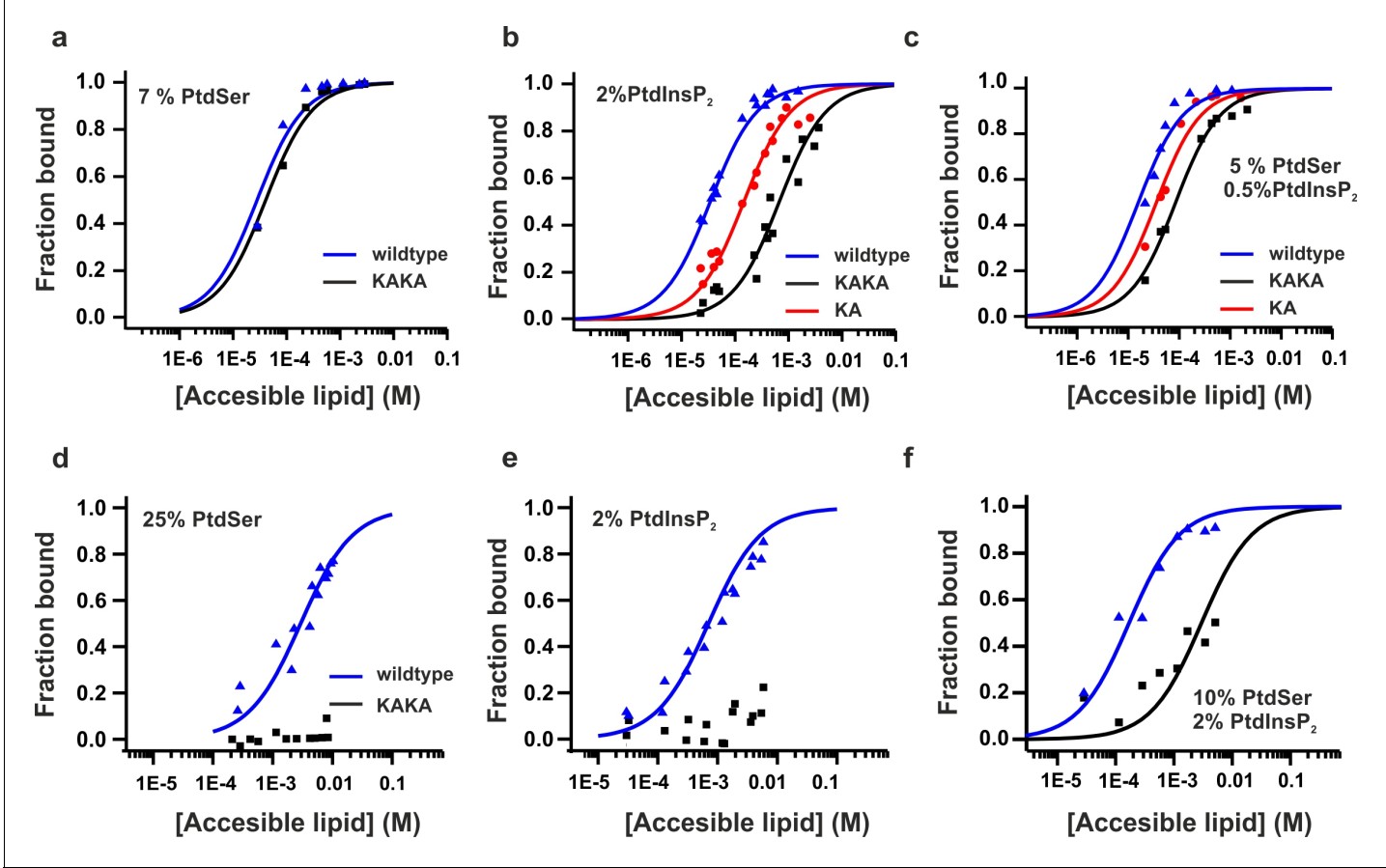

**Figure 5.** Equilibrium binding of the C2AB fragment and polybasic mutants to PtdChol, PtdSer and PtdIns(4,5)P$_2$ bilayers. (**a,b,c**) Ca$^{2+}$-dependent and (**d,e,f**) Ca$^{2+}$-independent partitioning of C2AB fragment into (**a,d**) PtdChol/PtdSer, (**b,e**) PtdChol/ PtdIns(4,5)P$_2$ and (**c,f**) PtdChol/PtdSer/ PtdIns(4,5)P$_2$ bilayers. (**d,e**) No binding of the KAKA mutant was observed in the presence of PtdSer or PtdIns(4,5)P$_2$ alone in absence of calcium . At equivalent charge densities, removal of lysine residues within the polybasic face reduced the membrane affinity in vesicles containing PtdSer and PtdIns (4,5)P$_2$ (**c**) more than it did in vesicles containing either (**a**) PtdSer or(**b**) PtdIns (4,5)P$_2$ in the presence of calcium. The lines represent fits to the data using Eq [1] (Materials and methods). Reciprocal molar partition coefficients obtained from data are listed in *Tables 2* and *3* (n = 2–3).

concentration of SNARE proteins is also high, so that significant competition with PtdIns(4,5)P$_2$ for syt-1 binding cannot be excluded. We conclude that the affinity of C2AB fragment to membranes containing PtdIns(4,5)P$_2$ and PtdSer is significant even in the absence of Ca$^{2+}$ (*Table 3*).

Taken together, our results demonstrate that PtdIns(4,5)P$_2$ primarily binds to the polybasic lysine patch of the C2B domain, which includes two Lys residues (K326 and K327), rather than to the Ca$^{2+}$-binding sites of C2B domain. As discussed below, the relative affinity of the C2AB fragment for PtdIns(4,5)P$_2$ versus its affinity for the SNAREs suggests that PI(4,5)P$_2$ is the likely target of the

**Table 2.** Reciprocal molar partition coefficients, *K*, for the Ca$^{2+}$-dependent membrane affinity of syt-1 C2 domains.

| Lipid composition | Syt1-C2AB WT | Syt1-C2AB K326A | Syt1-C2AB K326A/K327A |
|---|---|---|---|
| 7 mol % PtdSer | 1.0 (±0.1) x 10$^3$ | | 3.4 (±0.4) x 10$^2$ |
| 2 mol % PtdIns (4,5)P$_2$ | 2.2 (±0.2) x 10$^4$ | 5.1 (±0.5) x 10$^3$ | 1.1 (±0.1) x 10$^3$ |
| 5 mol % PtdSer + 0.5 mol % PtdIns(4,5)P$_2$ | 5.0 (±0.7) x 10$^4$ | 2.2 (±0.3) x 10$^4$ | 1.0 (±0.1) x 10$^4$ |

Values of *K* are expressed in units of M$^{-1}$. The errors represent the uncertainly obtained from the nonlinear regression used in fitting the data to *K*. The reciprocal of *K* corresponds to the accessible molar lipid concentration at which 50% of the protein is membrane bound (n = 2–3).

**Table 3.** Reciprocal molar partition coefficients, K, for the $Ca^{2+}$-independent membrane affinity of syt1 C2 domains.

| Lipid composition | Syt1–C2AB WT | Syt1–C2AB K326A/K327A |
|---|---|---|
| 25 mol % PtdSer | 2.7 ($\pm$0.3) x $10^2$ | not detected |
| 2 mol % PtdIns(4,5)P$_2$ | 1.1 ($\pm$0.2) x $10^3$ | not detected |
| 10 mol % PtdSer + 2 mol % PtdIns(4,5)P$_2$ | 4.7 ($\pm$0.9) x $10^4$ | 2.7 ($\pm$0.3) x $10^2$ |

Values of K are expressed in units of $M^{-1}$. The errors represent the uncertainly obtained from the nonlinear regression procedure used in fitting the data to K. The reciprocal of K corresponds to the accessible molar lipid concentration at which 50% of the protein is membrane bound (n = 2–3).

polybasic patch of the C2B domain, in agreement with recent reports (*Wang, 2016*; *Park et al., 2015*). Furthermore, $Ca^{2+}$ increases C2B's affinity for PtdIns(4,5)P$_2$, probably by neutralizing the negatively charged side chain at the $Ca^{2+}$-binding site of the C2B domain, allowing a tighter binding of the polybasic patch to PtdIns(4,5)P$_2$.

## Phosphoinosites and $Ca^{2+}$ decrease the dissociation rate of synaptotagmin-1

Although we focus on PtdIns(4,5)P$_2$ because it is the most abundant phosphoinositide in the plasma membrane (*Ueda, 2014*), it has been reported that syt-1 also binds to other phosphorylated phosphatidylinositols (PtdInsP$_x$) (*Vrljic et al., 2011*; *Wang et al., 2011*). Phosphatidylinositol phosphates are lipids that only differ from each other in the number of phosphate groups and/or the phospho site of the inositol ring, with different species being specifically associated with different intracellular membranes (*Chasserot-Golaz et al., 2010*). To gain insight into the binding of syt-1 to different phosphatidylinositols, we measured the binding kinetics of the C2AB fragment to vesicles containing 55% phosphatidylcholine (PtdChol), 22% phosphatidylethanolamine (PtdEth), 11% phosphatidylserine (PtdSer), 11% cholesterol (Chol) and 1% of different phosphatidylinositols (PtdInsP$_x$). Phosphorylated phosphatidylinositides increased the apparent affinities (*Figure 6a*), with the bimolecular association ($k_{on}$) and the unimolecular dissociation rate ($k_{off}$) constants being enhanced and reduced, respectively (*Figure 6b,c*). The 5–10-fold decrease in $k_{off}$ and $K_d$ ($k_{off}/k_{on}$) was progressively higher in the presence of bisphosphorylated phosphatidylinositols (PtdInsP$_2$) and triphosphorylated phosphatidylinositols (PtdInsP$_3$) than in the presence of monophosphorylated phosphatidylinositols (PtdInsP) (*Figure 6a,c*). While PtdInsP$_2$ or PtdInsP$_3$ increased the association rate constant moderately by 2–3-fold, PtdInsP or phosphatidylinositols (PtdIns) had no measurable effect on the binding kinetics (*Figure 6b*). Thus, these results suggest that syt-1 does not discriminate between the different tested phosphorylated phosphatidylinositol stereoisomers, and that the increase in affinity is mainly due to the higher charge density of the different PtdInsP$_x$ species. Our findings are in contrast to those of a recent study that came to a different conclusion based on a lipid overlay and a binding sensor assay (*Vrljic et al., 2011*). To further examine these conflicting results, we measured the binding of the head groups of different phosphatidylinositol phosphates to the C2B domain of syt-1 using ITC (*Figure 6d*). Surprisingly, Ins(1,3,5)P$_3$ showed lower affinity than the other tested stereoisomers (*Figure 6e*) because of a less favorable binding enthalpy (*Figure 6—figure supplement 1*).

Finally, we studied the binding kinetics of the C2AB fragment to vesicles containing PtdChol:PtdSer:PtdEth:Chol:PtdIns(4,5)P$_2$ (55:22:11:11:1 molar ratio) at four different $Ca^{2+}$concentrations (50, 100, 500 and 1000 μM). We were unable to detect binding at $Ca^{2+}$ concentrations lower than 50 μM. Nevertheless, for all other tested concentrations, the estimated bimolecular association rate constants ($k_{on}$) showed minor differences (*Figure 6g*), meaning that the increase in the $Ca^{2+}$-concentration did not drive the C2AB fragment to the membrane faster, at least not in this concentration regime. In contrast, the dissociation rate constant ($k_{off}$) showed a substantial decrease at concentrations over 50 μM (*Figure 6h*) with no further decrease at $Ca^{2+}$ > 100 μM. Because of this, the dissociation constant ($K_d$), followed a similar trend to the unimolecular dissociation rate constant ($k_{off}$) (*Figure 6f,h*).

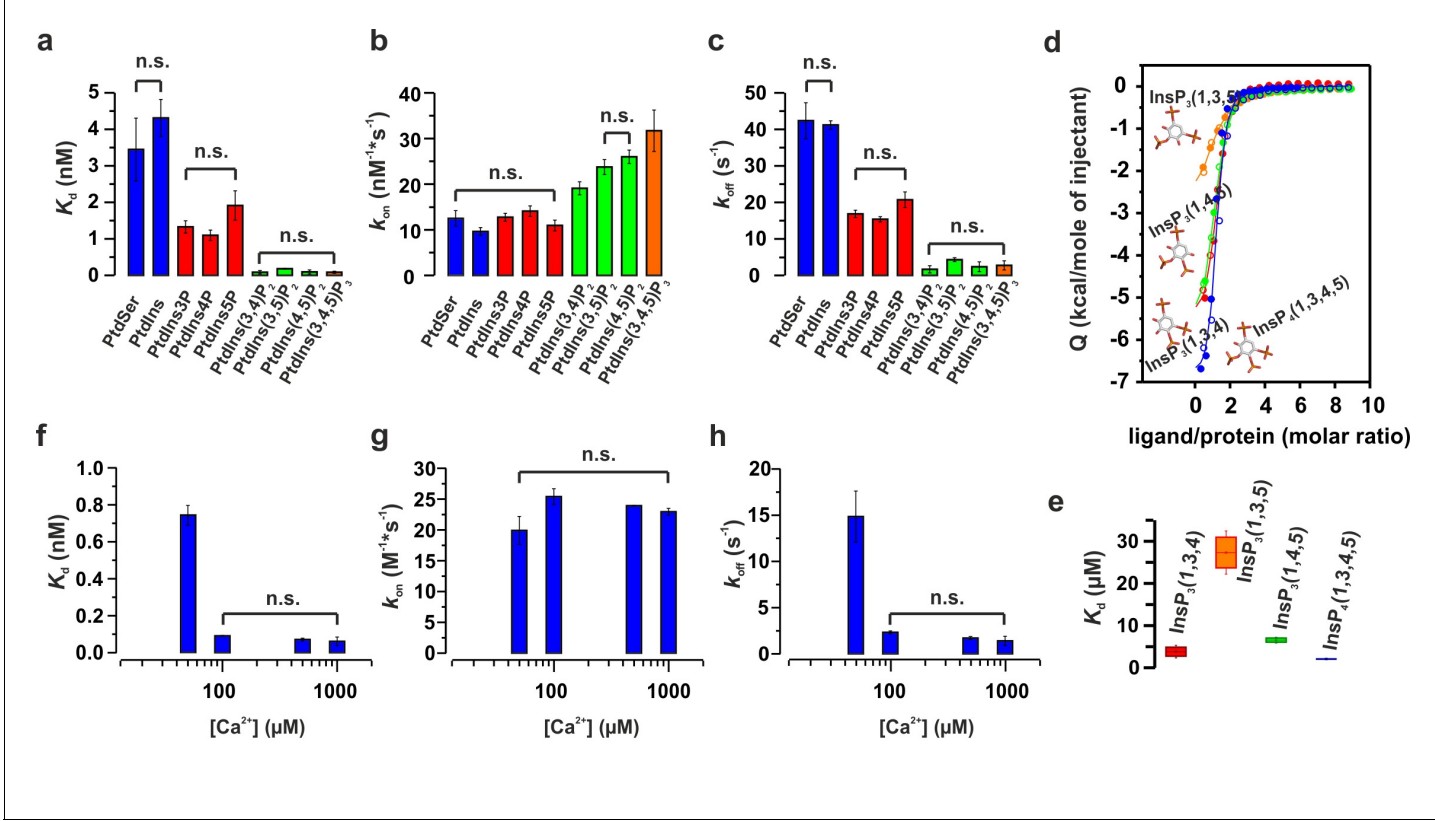

**Figure 6.** $Ca^{2+}$ and phosphoinositides increase the membrane affinity of syt-1 by decreasing the dissociation rate of the syt-1–membrane complex. (a) $K_d$, (b) $k_{on}$ and (c) $k_{off}$ calculated from stopped-flow experiments carried out by rapid mixing, as in **Figure 1**, of C2AB fragment with vesicles containing PtdChol/PtdSer/PtdEth/Chol/PtdInsP$_X$ (55:11:22:11:1 molar ratio), where X = 0–3 phosphate groups, in 20 mM HEPES (pH 7.4), 150 mM KCl, 1 mM EGTA and 1.1 mM $CaCl_2$ (100 μM free $Ca^{2+}$) at 37°C (n = 5–10). An increased number of phospho groups of phosphoinositides increases the affinity by decreasing the dissociation rate ($k_{off}$). A minor increase in $k_{on}$ was detected in the case of PtdInsP$_{2-3}$. (d) Thermodynamic characterization of the binding of C2B domain to head groups of physiological phosphoinositides by ITC. Titration of ~50 μM C2B domain (50 mM HEPES (pH 7.4), 150 mM NaCl and 1 mM $CaCl_2$) with main stereoisomers of InsP$_{3-4}$ at 25°C (n = 2).(e) Dissociation constants ($K_d$) calculated for experiments from d. (f) $K_d$, (g) $k_{on}$ and (h) $k_{off}$ calculated from stopped-flow experiments carried out at different $Ca^{2+}$ concentrations with vesicles containing PtdChol/PtdSer/PtdEth/Chol/PtdIns(4,5)P$_2$ (55:11:22:11:1 molar ratio) at 37°C (n = 5–10). $Ca^{2+}$ decreases the rate of dissociation drastically (h) and thereby increases the affinity (f). No significant difference was observed for the rate of association ($k_{on}$) (g).

The following figure supplement is available for figure 6:

**Figure supplement 1.** ITC parameters of the C2B domain binding to different isomers of InsP$_3$.

In summary, these results suggest that the association rate of the C2AB fragment is largely $Ca^{2+}$-independent. The increased membrane affinity of the C2AB fragment in the presence of $Ca^{2+}$ is due to a decreased dissociation rate from the membrane. As a result, $Ca^{2+}$ lengthens the period during which syt-1 remains bound once membrane binding has occurred.

## Discussion

In the present study, we used well-defined kinetic (stopped-flow) and structural (EPR and NMR) methods to better understand how the C2AB fragment of the vesicular $Ca^{2+}$ sensor synaptotagmin 1 (syt-1) binds to membranes. Three main conclusions can be derived from our data. First, PtdSer and PtdIns(4,5)P$_2$ act synergistically to promote deeper penetration of the $Ca^{2+}$-binding site of the C2B domain into the membrane. Second, $Ca^{2+}$ increases the affinity of syt-1 for PtdIns(4,5)P$_2$, which binds preferentially to the polybasic lysine patch of the C2B domain, and $Ca^{2+}$ acts by neutralizing

the negative charge of the $Ca^{2+}$-binding site. Third, $Ca^{2+}$ and phosphoinositides increase the affinity of syt-1 to membrane exclusively by reducing its dissociation rate.

It is well established that syt-1 binds to anionic phospholipids (mainly phosphatidylserine and phosphoinositides) via distinct regions, and that these interactions facilitate penetration of the syt-1 C2 domains into the membrane. However, neither the contribution of the individual lipids nor the conformational arrangement of the bound C2B domain was understood in molecular detail (*Kuo et al., 2009*; *van den Bogaart et al., 2012*; *Kuo et al., 2011*; *Radhakrishnan et al., 2009*; *Li et al., 2006*; *Bai et al., 2002*; *Schiavo et al., 1996*; *Araç et al., 2006*; *Davletov and Sudhof, 1993*; *Bai and Chapman, 2004*; *Bai et al., 2004*). Our results now show that PtdSer and PtdIns(4,5) $P_2$ act cooperatively by binding preferentially to different binding regions that influence each other. In the absence of $Ca^{2+}$, binding is mediated primarily by the interaction between the polybasic lysine patch and PtdIns(4,5)$P_2$, in which K326 and K327 play an important role as reported previously (*van den Bogaart et al., 2012*; *Bai et al., 2004*; *Araç et al., 2006*; *Li et al., 2006*).

In the absence of $Ca^{2+}$, the interaction of syt-1 with the bilayer is limited to an interfacial absorption of the C2B domain via its polybasic region, where membrane insertion of C2B $Ca^{2+}$-binding loops is prevented by charge repulsion between the negatively charged $Ca^{2+}$-binding site and the negatively charged membrane interface (*Bai et al., 2004*; *Kuo et al., 2009*). Intriguingly, the negatively charged binding site of the C2B domain also reduces the affinity of this domain for PtdIns(4,5) $P_2$ by electrostatic repulsion (see model in *Figure 7*, left). In the presence of $Ca^{2+}$, several effects operate synergistically to increase membrane affinity. First, $Ca^{2+}$ binding polarizes the $Ca^{2+}$-binding loops of the C2B domain and PtdSer completes the coordination sphere of the bound $Ca^{2+}$, allowing syt-1 to penetrate into the membrane. Second, the neutralization of charge at the $Ca^{2+}$ binding site of the C2B domain reduces electrostatic repulsion at the polybasic region and promotes the association of PtdIns(4,5)$P_2$ (*Figure 7*, right). This cross-talk between the two sites explains the conflicting conclusions regarding the role of the $Ca^{2+}$-binding sites and the polybasic lysine patch in binding to anionic membrane lipids. The $Ca^{2+}$-binding sites preferentially bind to PtdSer, for which they form a defined binding pocket (*Honigmann et al., 2013*) but not to phosphoinositides, in contrast to previous suggestions (*Vrljic et al., 2011*; *Zhang et al., 1998*; *van den Bogaart et al., 2012*). Calcium increases the affinity of syt-1 towards PtdIns(4,5)$P_2$ simply by neutralizing negative charge at the $Ca^{2+}$-binding site. It should be noted that by using $InsP_3$, the soluble head group of PtdIns(4,5)$P_2$, there is no membrane interface in this experiment and no charge repulsion between the C2AB fragment and liposomes containing PtdIns(4,5)$P_2$. This fact could explain the non-specific interactions with the $Ca^{2+}$-binding sites (*Figure 3b,c*) and the weak effect of the presence of the SNARE complex on $InsP_3$–C2AB fragment binding (*Figure 4g,i*).

The question then arises of how the binding between the C2AB domain and the SNARE complex fits into the picture. In particular, we need to clarify whether there is synergism or competition between syt binding to SNAREs and syt binding to acidic membrane lipids during the triggering of exocytosis. Clearly, the polybasic patch of the C2B domain is involved in SNARE binding with high affinity as shown earlier (*Brewer et al., 2015*; *Zhou et al., 2013*) and as confirmed by our ITC data. Moreover, mutations that impair syt–SNARE binding also alter syt-membrane binding; in addition, they alter the triggering of exocytosis when the corresponding mutants are introduced into neurons (*Brewer et al., 2015*; *Wang et al., 2016*). However, the available evidence suggests that the polybasic region preferentially interacts with PtdIns(4,5)$P_2$ rather than with SNAREs under normal ionic conditions (*Park et al., 2015*; *Wang et al., 2016*). For example, pulse EPR and fluorescence cross-correlation spectroscopy indicate that the Syt–PtdIns(4,5)$P_2$ interaction persists under conditions that eliminate the syt–SNARE interaction (*Park et al., 2015*), and the data presented here in *Figure 4g,i* show that $InsP_3$ binding to the C2AB fragment is only slightly affected by the presence of the SNAREs. Nevertheless, the syt–SNARE interaction is conformationally heterogeneous and quite dynamic, and sites other than the polybasic patch may interact with the SNAREs in the presence of PtdIns(4,5)$P_2$. This may result in a more complex interplay between PtdIns(4,5)$P_2$, SNARE proteins and syt-1 under physiological conditions (*Wang et al., 2016*; *Zhou et al., 2015*; *Park et al., 2015*; *Brewer et al., 2015*).

Despite the clear evidence for high-affinity syt–SNARE binding in the test tube, we favor the view that syt-phospholipid binding is the main component in the triggering mechanism of syt-1, with the SNARE binding either being physiologically irrelevant or being synergistic at best. For instance, all models trying to picture synaptotagmin complexes with partially zippered SNARE complexes face

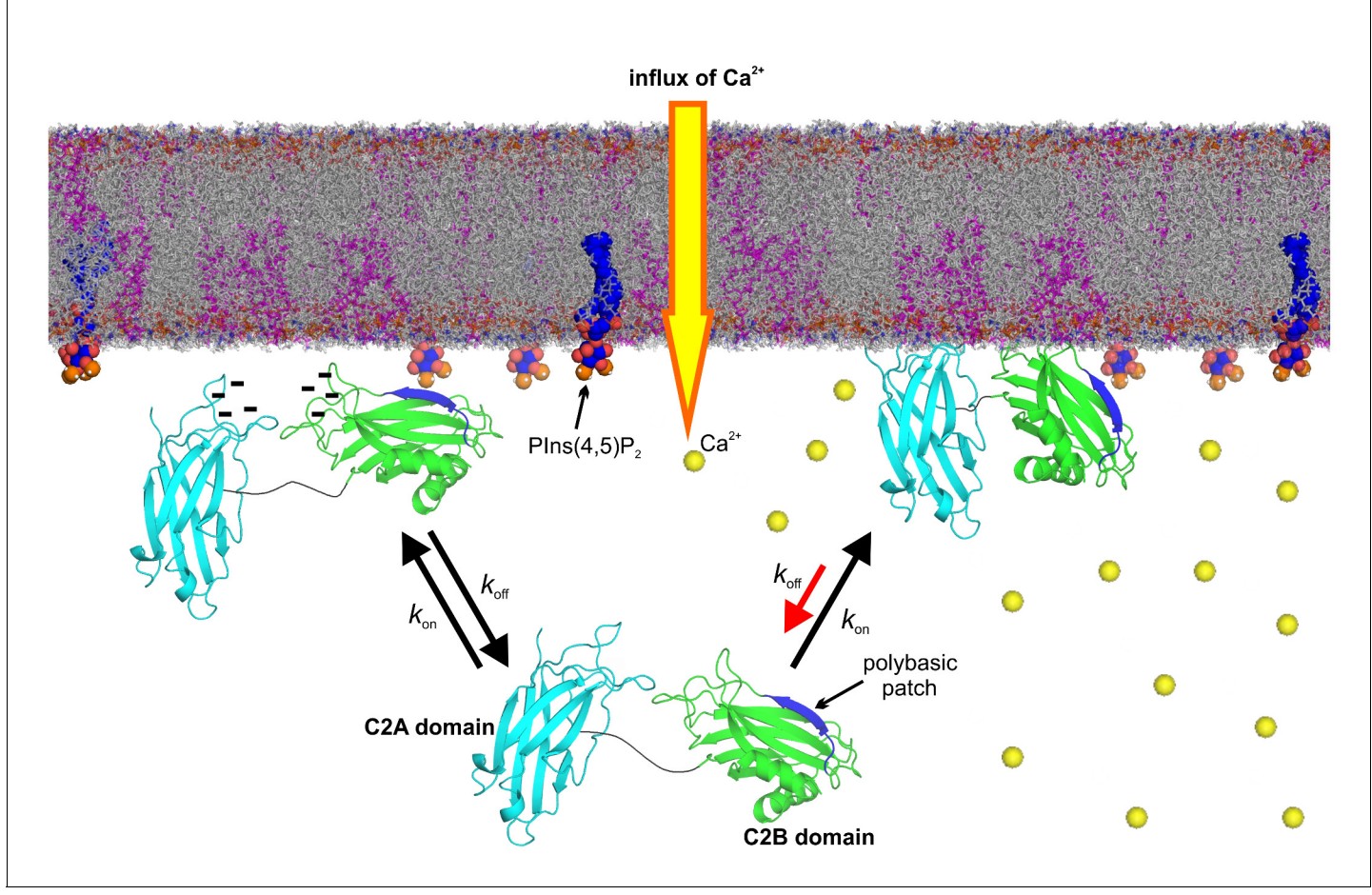

**Figure 7.** Model of the membrane-binding mechanism of syt-1. In the absence of Ca$^{2+}$, syt-1 is attached to the presynaptic membrane interface. Syt-1 binds to PtdIns(4,5)P$_2$ through its C2B polybasic patch (*Bai et al., 2004*; *Kuo et al., 2009*) in transient encounters, but the negative charges of the Ca$^{2+}$-binding pockets prevent penetration of the C2 domains into the presynaptic membrane because of the electrostatic repulsion between them, leading to a high rate of dissociation. Upon Ca$^{2+}$ influx, Ca$^{2+}$ binding neutralizes the negative charge of the Ca$^{2+}$-binding sites. As a consequence, phosphatidylserine completes the sphere of coordination of Ca$^{2+}$ and allows the insertion of the hydrophobic residues at the tips of the C2 domains. Simultaneously, the polybasic patch enhances its affinity to phosphoinositides, leading to deeper penetration of the C2B Ca$^{2+}$-binding site into the bilayer. Together, these events decrease the rate of dissociation of syt-1 from the membrane and enhance its penetration into the core of the presynaptic membrane, which eventually leads to SNARE-mediated membrane fusion. Nuclear magnetic resonance structures of the C2A domain (PDB 1BYN [*Shao et al., 1998*]) and C2B domain (1K5W [*Fernandez et al., 2001*]) of syt-1 and a molecular dynamic membrane simulation were rendered using PyMOL Molecular Graphics System (Schrödinger,LLC, http://www.pymol.org). The membrane used in this illustration was generated using the Membrane Builder input generator module in CHARMM-GUI (*Jo et al., 2008*) for a bilayer with PtdChol:PhdSer:PtdEth:Chol:PtdInsP$_2$ at a molar ratio of 55:11:22:11:1.

serious spatial challenges when trying to fit them into a 3D-model between the vesicle and plasma membrane, particularly when considering that both Munc18 and Munc13 (*Ma et al., 2013*) (or even the larger CAPS proteins [*Daily et al., 2010*]) are thought to be bound to such activated SNARE complexes. Moreover, synaptotagmins or other C2 domain proteins are not required for the basic and evolutionarily conserved SNARE engine. They do not participate in other intracellular fusion reactions and are lacking altogether in more primitive eukaryotic cells such as yeast (*Craxton, 2010*). Moreover, the canonical function of Ca$^{2+}$-binding C2 domains appears to be binding to acidic membrane lipids (*Corbalan-Garcia and Gomez-Fernandez, 2014*), see also http://scop.mrc-lmb.cam.ac. uk/scop/. This is true for soluble proteins whose Ca$^{2+}$-dependent translocation to plasma membranes is mediated by C2 domains (classical examples include protein kinase C and phospholipase A), and also for transmembrane proteins owning multiple tandem C2 domains, such as ferlins and

extended-synaptotagmins (see e.g *Idevall-Hagren et al. (2015)*; *Fernandez-Busnadiego et al. (2015)*; *Pangršič et al. (2012)*).

In summary, a picture is emerging in which syt-1 continuously probes the membrane with high frequency both in the $Ca^{2+}$-bound and -unbound state. Binding of $Ca^{2+}$ increases the dwell time of syt-1 on the membrane by allowing more intimate contact with the bilayer rather than by promoting an enhanced synaptotagmin–membrane association. The model that we propose in *Figure 7* is in accordance with data from functional experiments carried out in intact synapses. For instance, mutations in the C2B domain polybasic lysine patch reduce the amplitude of the fast component of EPSC in excitatory neurons (*Li et al., 2006*). Moreover, InsP$_6$ (which binds to the syt-1 polybasic lysine patch [*Joung et al., 2012*]) suppresses autaptic EPSCs via the C2B domain of syt-1 (*Yang et al., 2012*) and weakens the $Ca^{2+}$-dependent binding of syt-1 to membranes (*Lu et al., 2002*). Thus, the docking/priming function of the polybasic lysine patch facilitates release but is not essential since the $Ca^{2+}$-binding sites remain intact and still allow for binding to PtdSer and membrane insertion. Indeed, increasing PtdSer concentrations increase the frequency of fusion events in PC12 cells (*Zhang et al., 2009*). Furthermore, mutated proteins in which the hydrophobicity of the $Ca^{2+}$-binding loops of syt-1 C2 domains were increased enhanced both $Ca^{2+}$-dependent membrane binding and neurotransmitter release in autaptic hippocampal cultures (*Rhee et al., 2005*). Additionally, a recent report determining the ultrastructure of the mouse hippocampal organotypic culture revealed a reduction of synaptic vesicles within 0–5 nm of the active zone in syt-1 KO synapses. This report supports our model, in which syt-1 continuously probes the plasma membrane in the latest steps of the docking/priming process (*Imig et al., 2014*).

Despite all of this detailed knowledge, we still do not know how the synaptotagmin–membrane interaction triggers neurotransmitter release. While, as discussed above, the role of SNARE binding is controversial (*Bacaj et al., 2015*; *Zhou et al., 2015*; *Brewer et al., 2015*; *Park et al., 2015*), it is largely accepted that $Ca^{2+}$-enhanced binding to acidic membrane lipids is crucial for triggering exocytosis. Our data suggest that two features of the syt-1–membrane interaction play a crucial role: (i) the membrane insertion of the $Ca^{2+}$-binding loops rather than an exclusively electrostatic membrane adsorption, and (ii) an increased dwell time of the protein on the membrane surface resulting from an elevated kinetic stability of the syt-1–membrane complex. In particular, the latter finding was unexpected. It is conceivable that the increased dwell time brings the membranes closer together and thus triggers the firing of SNARE complexes. It is also conceivable that the deeper insertion into the membrane destabilizes the bilayer at the prefusion contact site and thus helps to overcome the energy barrier towards fusion. Hopefully, our findings will trigger novel approaches towards elucidating the molecular mechanism of the still enigmatic triggering event in neuronal exocytosis.

## Materials and methods

### Liposomes

For stopped-flow measurements, lipid were purchased from Avanti Polar Lipids, except for dansyl-labeled phosphatidylethanolamine (dansyl-DHPE), which was purchased from Invitrogen. Lipids were mixed in appropriate amounts and dried under a stream of nitrogen, and traces of organic solvent were removed under vacuum for at least 3 hr. Lipid mixtures were resuspended in 20 mM HEPES, 150 mM KCl, 1.1 mM CaCl$_2$ and 1 mM EGTA (for 100 µM free $Ca^{2+}$concentration calculated using http://maxchelator.stanford.edu) at pH 7.4. Large unilamellar vesicles of around ~100 nm were prepared by extrusion as described earlier (*Arbuzova et al., 1997*). Total phospholipid concentration was determined using the total phosphate determination method (*Böttcher et al., 1961*). For vesicle sedimentation assay and EPR measurements, 1-palmitoyl-2-oleoyl-sn-glycero-3-phosphatidycholine (POPC), 1-palmitoyl-2-oleoyl-sn-glycero-3-phosphatidyserine (POPS), and L-α-phosphatidylinositol-4,5-bisphosphate (ammonium salt) (PtdIns(4,5)P$_2$) were purchased from Avanti Polar Lipids (Alabaster, AL), to make large unilamellar vesicles (LUVs) of 100 mM concentration at the following molar ratios: PtdChol/PtdSer (3:1 and 4:1, molar ratio), PtdChol/PtdIns(4,5)P$_2$ (98:2 and 95:5, molar ratio), PtdChol/PtdSer/PtdIns(4,5)P$_2$ (87.5:10:2.5, molar ratio). The lipids were prepared as described previously (*Kuo et al., 2011*). The dried lipid film was re-suspended in either sucrose buffer (176 mM sucrose, 1 mM MOPS, pH 7.0) or $Ca^{2+}$ buffer (1 mM CaCl$_2$, 1 mM MOPS, 100 mM KCl, pH 7.0), and

vesicles were formed using freeze-thaw cycles and extrusion as previously described (*Kuo et al., 2011*). To produce sucrose-loaded LUVs, another wash step with $Ca^{2+}$ buffer was used to remove the external sucrose solution followed by ultracentrifugation at 160,000 × g for 1 hr.

## Protein constructs and purification

All synaptotagmin-1 proteins were derived from *Rattus norvegicus* and expressed as pET28a constructs, namely the isolated C2A domain (aa 97–273), the C2B domain (aa 262–421), the C2AB fragment of synaptotagmin-1 (aa 97–421) and $Ca^{2+}$-binding mutants of the soluble domain: C2a*B (D178A, D230A, and D232A), C2Ab* (D309A, D363A, and D365A), and C2a*b* (D178A, D230A, D232A, D309A, D363A, and D365A), have been described before (*Stein et al., 2007*). The KAKA mutant (K326A, K327A) has also been described earlier (*Radhakrishnan et al., 2009*). SNARE proteins were composed of syntaxin 1A (residues 188–259), SNAP-25 (residues 1–206) and synaptobrevin (residues 1–96).

Proteins were expressed in *Escherichia coli* strain BL21 (DE3) and purified using $Ni^{2+}$-nitrilotriacetic acid beads (Qiagen GmbH) followed by ion exchange and gel filtration chromatography on the ÄKTA system (GE Healthcare) as described earlier (*Radhakrishnan et al., 2009*; *Fasshauer et al., 2002*). SNARE-complex assembly was performed as previously described (*Fasshauer et al., 2002*) by mixing the SNARE proteins and incubating them overnight at 4°C. SNARE-complex was purified by ion exchange and gel filtration chromatography, followed by concentration with a 30 kDa cut-off concentrator (Sartorius Stedim Biotech GmbH, Göttingen, Germany).

For EPR, NMR and vesicle sedimentation assay, DNA for rat syt-1 (P21707) was obtained from Dr. Carl Creutz (Pharmacology Department, University of Virginia). In the pGEX-KG, vector encoding amino acid residues 96–421 and 249–421 (syt-1 C2B) (*Damer and Creutz, 1994*; *Kuo et al., 2009*) were used to produce constructs 136–260 (syt-1 C2A) and 136–421 (syt1 C2AB) by ligation into the plasmid vector pGEX-KG following the coding region for GST as described previously (*Bhowmik et al., 2008*). The single native cysteine residue at position 277 was mutated to alanine and single and double alanine mutants (K326A and K326A/K327A) were produced in the C2AB construct (136–421) by QuickChange site-directed mutagenesis (Agilent, Santa Clara, CA). Single cysteine substitutions were also produced to create C2AB mutants of M173C, V304C, L323C, K327C and T329C. All mutagenesis was confirmed by DNA sequencing.

The expression and purification of syt-1 C2A, C2B, and C2AB constructs were carried out as previously published (*Kuo et al., 2009*; *Herrick et al., 2009*). The wild-type and mutant plasmids were expressed in BL21 (DE3) pLysS cells (Invitrogen, Grand Island, NY), and grown in LB media. The protein was purified using a GST affinity column followed by a second purification step using ion exchange chromatography. An SP column was used for the C2AB fragment and C2B domain, and a Q column was used for the C2A domain to remove any remaining contaminants, such as nucleic acids. Syt-1 C2AB prepared in this manner is correctly folded as indicated by CD (*Herrick et al., 2006*) and NMR. This C2AB fragment is also found to bind membranes in a $Ca^{2+}$-dependent manner. SDS Page indicated that the proteins were pure with appropriate molecular weights of 17.4 for C2A domain, 20.8 for C2B domain, and 34.5 for C2AB fragment. The UV absorbance at 278 nm was used to insure that protein fractions were free of nucleic acid contaminant. The protein concentrations were determined using a Bradford Protein Assay (Thermo Fisher Scientific, Rockford, IL).

## Isothermal titration calorimetry (ITC)

ITC experiments were carried out as previously described (*Joung et al., 2012*). Titration of ~50 µM protein (see figure legend) with different ligands were performed at 25°C in 50 mM HEPES (pH 7.4) and 150 mM NaCl, in the presence or absence of 1 mM $CaCl_2$. Buffer without calcium was prepared as reported (*Radhakrishnan et al., 2009*) and checked with rhod-5N (Invitrogen). To obtain the effective heat of binding, results of the titration were corrected using buffer-protein and ligand-buffer controls. Finally, the NITPIC (*Keller et al., 2012*) and the Origin9 software (Origin Labs Inc.) were used to analyze these data.

1,2-hexanoyl-sn-glycero-3-phospho-L-serine ($C_6$-PtdSer) was titrated under critical micellar concentration (CMC) to avoid the formation of $C_6$-PtdSer micelles (http://www.avantilipids.com) that interfere in the titration.

## Kinetic fluorescence experiments

Kinetic experiments were carried out on an Applied Photophysics SX.20 stopped-flow spectrophotometer (Applied Photophysics, Surrey, UK) in 20 mM HEPES (pH 7.4), 150 mM KCl, and 1 mM EGTA and different free $Ca^{2+}$ concentrations (calculated using http://maxchelator.stanford.edu), as described previously (*Hui et al., 2005*). FRET was used to monitor the time course of the C2AB fragment vesicle binding. The excitation wavelength was set at 280 nm and a 470 nm cut-off filter was used to collect the dansyl emission for the different vesicle concentrations tested. The resulting time courses were fit to a single-exponential function:

$$F(t) = F_0 + A_{obs}{}^* e^{(-k_{obs}*t)}$$

where F(t) equals the observed fluorescence at time t, $F_0$ is a fluorescence offset representing the final fluorescence, $A_{obs}$ equals the amplitude, and $k_{obs}$ is the observed rate constant. Observed rate constants were plotted as a function of lipid vesicle concentration (v), calculated assuming 90,000 phospholipid molecules per vesicle (*Arbuzova et al., 1997*), and fitted with the equation:

$$k_{obs} = k_{on}[v] + k_{off}$$

where $k_{on}$ represents the apparent association constant, and $k_{off}$ the apparent dissociation rate constant. The ratio of $k_{off}$ to $k_{on}$ provides the calculated apparent vesicle dissociation constant ($K_d$).

## Vesicle sedimentation assay

An ultracentrifugation technique (*Buser and Mclaughlin, 1998*) was used to measure the equilibrium membrane binding of C2AB and C2AB mutants to membranes of varied lipid composition. For experiments where tryptophan emission was used to detect C2AB, final protein concentrations of 0.2–0.35 μM were used with sucrose-loaded LUVs at lipid concentrations ranging from 0.02 mM to 15 mM. The vesicles and C2AB were incubated at room temperature for 10 min, followed by centrifugation at 160,000 × g for 1 hr to pellet the LUVs. The concentration of C2AB in the supernatant was determined and used to calculate the fraction of membrane-bound protein ($f_b$). A phosphate assay (*Ames, 1966*) was used to determine the final lipid concentrations. BODIPY-maleimide labeled C2AB was used in cases were binding affinities were high, in order to maintain sufficiently low protein-to-lipid ratios (*Buser and Mclaughlin, 1998*). For PtdChol: PtdSer LUVs, accurate binding affinities were achieved at lipid to protein molar ratios of 140:1 or higher. For each experimental condition, at least two measurements of the fraction of bound protein, $f_b$, were made to determine a reciprocal molar partition coefficient, K. This partition coefficient represents the accessible lipid concentration, [L], where half the protein is bound and is given by:

$$f_b = \frac{K[L]}{1 + K[L]} \tag{1}$$

The value of K ($M^{-1}$) and the standard errors were determined from the data using the fitting function in OriginPro 7.5 (Origin Lab, Northampton, MA).

## Proton-nitrogen HSQC NMR

C2A and C2B were expressed in BL21 (DE3) pLysS cells, and grown in minimal media where $^{15}NH_4Cl$ (Cambridge Isotopes, Andover, MA) was the sole nitrogen source. The purification followed the same protocol mentioned above. The final protein at a concentration of 0.4–0.8 mM in NMR buffer (3 mM CaCl2, 150 mM NaCl, 50 mM MES, pH 6.3) was used for two-dimensional 1 hr $^{15}N$ HSQC experiments (*Ubach et al., 2001*). The ligands (o-phosphoserine or InsP₃) were titrated into C2A or C2B at concentrations ranging from 0.05 mM to 4 mM. The final NMR protein samples contained 10% $D_2O$ and were placed in Shigemi tubes (Sigma-Aldrich, St. Louis, MO). The experiments were conducted in Bruker 600 MHz NMR spectrometer at temperature 27°C, and DDS (4,4-dimethyl-4-silapentane-1-sulfonic acid) in NMR buffer was used as a chemical shift standard. The NMR data were processed in NMRPipe (*Delaglio et al., 1995*) and the residue assignments were matched in Sparky (*Goddard and Kneller, 2008*) from the PDB NMR resonance assignments of C2A (PDB 1BYN) and C2B (PDB 1K5W) (*Fernandez et al., 2001*; *Shao et al., 1997*; *Shao et al., 1998*).

## EPR measurements

The C2AB mutants, M173C, V304C, L323C, K327C, T329C, were purified and spin labeled using MTSL (1-oxy-2,2,5,6-tetramethylpyrroline-3-methyl; Santa Cruz Biotech, Dallas, TX). Each sample was spin labeled at a 1:1:10 mole ratio of protein:DTT:MTSL, and the mixture was passed through a HiPrep 26/10 desalting column (GE Healthcare, Pittsburgh, PA) to remove any free spin labels. The spin-labeled protein was concentrated to 50–200 uM using an Amicon centrifugal concentrator (EMD Millipore, Billerica, MA).

EPR spectra and power saturation measurements were performed on an X-band Bruker EMX spectrometer using a room temperature ER 4123D dielectric resonator. Samples in the $Ca^{2+}$-free condition contained 5 mM EGTA. The samples with LUVs had a final lipid concentration of 25 mM with a lipid-to-protein ratio of at least 200:1, ensuring that all protein was completely membrane associated. All spectra were 100 Gauss scans recorded at 2 mM incident microwave power and 1 G modulation. The spectra were baseline corrected and normalized by double integration using a Lab-VIEW (Austin, TX)-based program (Dr. Altenbach, UCLA).

Continuous wave power saturation experiments were performed as previously described (*Kuo et al., 2011*), using 12 microwave power steps ranging from 0.6 to 36 mW. A LabVIEW-based program (Dr. Altenbach, UCLA) was used to plot the data points and determine P1/2 values from which ΔP1/2 values for NiEDDA and $O_2$ were obtained. These data yielded depth parameters, Φ, which were used to estimate the distance (x) from the lipid phospholipid phosphates using an empirically derived calibration curve (*Frazier et al., 2002*).

## Membrane docking of the Syt1 C2B domain

Membrane depth data obtained from EPR power saturation were used to generate models for the position of the C2B domain in bilayers containing either PtdSer or PtdIns(4,5)$P_2$. The approach utilized a version of Xplor-NIH (*Schwieters et al., 2003*) that included a plane distance potential, and followed the general procedure described previously (*Herrick et al., 2009*). Briefly, to the high-resolution NMR structure for the C2B domain (PDB ID: 1K5W [*Fernandez et al., 2001*]) the site-scan feature of the software package MMM (*Polyhach et al., 2011*; *Polyhach and Jeschke, 2010*) was used in conjunction with the warsh rotameric library to determine possible rotamers for the positions of interest. The most probable rotamer was then attached to each site. The depth parameters shown in *Table 1* were used as point-to-plane distance restraints and the labeled structure was subjected to simulated annealing and energy minimization to dock the domain to the membrane interface. The spin label atoms and all protein backbone atoms were fixed for the duration of the simulated annealing runs. For these point-to-plane restraints, the distance range was set to the uncertainty in the label position, which was determined from the experimental error in the membrane depth parameter and the empirically derived calibration curve (*Herrick et al., 2006*). Approximately 100 structures were generated from each simulated annealing run. The structures were analyzed and figures were generated using the program PyMOL (Schrödinger, Cambridge, MA).

## Statistics

All data are shown as means ± SD (n = 2–10), and all statistical analyses were performed by one-way ANOVA and Fisher post hoc test (p-value < 0.05).

## Acknowledgements

We thank U Ries for protein purification assistance and Prof. MV Rodnina, for access to the stopped-flow facilities. We thank Dr. JM Hernández for critical review and Dr. Milon Pohl and Dr. Wolf Holtkamp for technical assistance. We thank all members Jahn's laboratory for discussion and comments.

## Additional information

### Competing interests

RJ: Reviewing editor, *eLife*. The other authors declare that no competing interests exist.

## Funding

| Funder | Grant reference number | Author |
| --- | --- | --- |
| National Institutes of Health | P01 GM072694 | David S Cafiso<br>Reinhard Jahn |
| Deutsche Forschungsge-meinschaft | SFB803 | Ángel Pérez-Lara<br>Partho Halder<br>Reinhard Jahn |
| Max-Planck-Gesellschaft | Postdoctoral Fellowship | Ángel Pérez-Lara<br>Partho Halder |

The funders had no role in study design, data collection and interpretation, or the decision to submit the work for publication.

## Author contributions

ÁP-L, Conception and design, Acquisition of data, Analysis and interpretation of data, Drafting or revising the article; AT, SBN, DAN, Acquisition of data, Analysis and interpretation of data, Drafting or revising the article; PH, Conception and design, Drafting or revising the article; MT, Acquisition of data, Drafting or revising the article; KT, Analysis and interpretation of data, Drafting or revising the article; DSC, RJ, Conception and design, Analysis and interpretation of data, Drafting or revising the article

## Author ORCIDs

Ángel Pérez-Lara, http://orcid.org/0000-0002-2736-3501
Reinhard Jahn, http://orcid.org/0000-0003-1542-3498

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
