## [Decision Letter]

Thank you for submitting your article "PtdInsP_2_ and PtdSer cooperate to trap synaptotagmin-1 to the plasma membrane in the presence of calcium" for consideration by *eLife*. Your article has been reviewed by three peer reviewers, and the evaluation has been overseen by a Reviewing Editor and Gary Westbrook as the Senior Editor. The following individuals involved in review of your submission have agreed to reveal their identity: Jose Rizo-Rey.

The reviewers have discussed the reviews with one another and the Reviewing Editor has drafted this decision to help you prepare a revised submission.

Summary:

This manuscript re-examines how the C2 domains of synaptotagmin-1 bind to membranes. This is a subject that has been investigated extensively for many years, but the authors make a good case that, despite so many studies, we still do not have a clear picture of how negatively charged phospholipids such as PS and phosphoinositide phosphates (particularly PIP2) cooperate (or not) in binding to synaptotagmin-1. This is an area of major interest because it is well established that both calcium-independent and calcium-dependent membrane binding is critical for the role of synaptotagmin-1 as a calcium sensor in fast neurotransmitter release. The results of the paper show that PS and PIP2 synergize to promote synaptotagmin-1 binding in the absence of calcium and, upon calcium binding, enhance the insertion of the C2 domain calcium-binding loops into the lipid bilayer, that calcium enhance PIP2 binding by making the electrostatic potential more positive, and that the enhancements of the affinity of synaptotagmin-1 for membranes caused by phosphoinositides arise from reductions in the dissociation rates.

For the non-specialist reader, it is difficult to determine the advances that have been made in this study because previous publications came to related conclusions, especially upon casual reading. Thus, the manuscript would benefit greatly by describing previous results (see for a partial list below) in a balanced fashion and then placing the results of this study into historic perspective. The reviewers also raised a number of substantive issues outlined below, and additional experiments are suggested to strengthen studies of the polybasic region, and to reconcile their results with another recently published study (Wang et al. *eLife* 2016).

Major comments:

1) The authors conclude that PIP2 binds to a specific binding site of the C2B domain. However, their NMR data (Figure 3) is in apparent contradiction with this conclusion. It is unlikely that the observed chemical shift changes (which span residues more than 30 angstroms apart) can be explained by PIP2 binding to a single site. The NMR data are much more easily explained by a model whereby PIP2 can quickly switch between different binding modes along the polybasic region (which includes six basic residues along a single β-strand plus additional basic residues in nearby strands), and also the calcium-binding region. This notion does not contradict the approximately 1:1 stoichiometry measured by the authors, as one single molecule of PIP2 could move around the polybasic region and its strong negative potential would hinder binding of a second PIP2 molecule to the region. The overall lack of a specific, defined binding site is supported by the lack of specificity for different stereoisomers of phosphatidylinositols shown by the authors (subsection “Phosphoinosites and Ca^2+^ decrease dissociation rate of synaptotagmin-1”, first paragraph).

2) The authors only mutate two residues (K326 and K327) among the many that form the polybasic region. Without comparing the effects of mutations in all the residues of the polybasic region, it is difficult to draw conclusions about specificity for a single, well-defined binding site. Studies with additional mutations would strengthen this part of the manuscript. And one should also consider that the side chains of K326 and K327 are on opposite sides of a β strand.

3) In the second paragraph of the Discussion, the authors state that the calcium binding sites specifically bind PS but not phosphoinositides. However, it has been known for a long time that other negatively charged phospholipids such as π can bind similarly to the synaptotagmin-1 C2 domains; hence, the specificity for PS is quite relative. Note also that the NMR data (Figure 3) show that there is some degree of binding of PIP2 to the calcium-binding region, even though it is less populated than binding to the polybasic region.

4) In the fourth paragraph of the subsection “PtdInsP_2_ binding to the C2B domain polybasic patch”, the authors describe ITC experiments showing that the presence of the SNARE complex does not alter binding to the soluble InsP_3_ (Figure 4), and conclude that this result supports the notion that synaptotagmin-1 binds to PIP2 preferentially over the SNARE complex, in contrast to conclusions drawn in Brewer et al., 2015, and also in contrast to the data presented in a recent paper (Wang S, Li Y, Ma C (2016) *eLife* 5:1689-1699). However, the conclusion from the authors was obtained with the soluble head group whereas Brewer et al., 2015 used liposomes containing 1% PIP2. In Radhakrishnan et al., 2009 it has been shown that binding of synaptotagmin-1 to liposomes containing 1% PIP2 is very weak (Radhakrishnan et al., 2009), and in this current paper the experiments performed with liposomes containing 2.5% PIP2 (no PS) also seem to suggest a very low affinity (subsection “PtdSer and PtdIns(4,5)P_2_ play a synergistic role in membrane binding of synaptotagmin-1”, second paragraph and Figure 1). Hence, these experiments with liposomes correlate better with the conclusions of ref. 25 than with the ITC results. It is also important to emphasize that the SNARE complex aggregates at the concentrations used in the ITC experiments (50 μM), which could influence the results obtained. The authors should discuss these issues and consider whether ITC experiments performed with InsP_3_ in the presence of the SNARE complex are conclusive, also in view of the recent paper by Wang, Li, and Ma (2016). Additional experiments may be needed to reconcile these different conclusions.

5) The interpretation of the results in terms of static pictures is only a model. It is possible, or even likely that the interactions of the synaptotagmin-1 C2 domains with membranes are dynamic, and such an interpretation is perfectly compatible with the data presented by the authors in this paper.

6) As mentioned above, the work lacks discussion of previous work, including (this is not meant to be a comprehensive list):

A) Bai J, Tucker WC, Chapman ER (2004) Nat Struct Mol Biol 11:36-44 which shows that the polybasic region of the Syt1 C2B domain interacts with PIP2 containing membranes without and with calcium.

B) Bai J, Wang P, Chapman ER (2002) Proc Natl Acad Sci U S A 99:1665-1670 which finds that the calcium binding loops of the Syt1 C2B domain insert into the membrane upon calcium binding.

C) Rufener E, Frazier AA, Wieser CM, et al. (2005) Biochemistry 44:18-28 which determined the orientation and position of the Syt1 C2B domain upon calcium binding into the membrane using EPR.

D) Van Bogaart G Den, Meyenberg K, Diederichsen U, Jahn R (2012) J Biol Chem 287:16447-16453 which shows that the polybasic patch of tagmin is critical in the protein's ability to bind PIP2.

E) Zhou Q, Lai Y, Bacaj T, et al. (2015) Nature 525:62-67 which revealed interfaces between synaptotagmin-1 and SNARE complex, one of which was found to be very important for function in the neuron, and suggested a model of a pre-assembled synaptotagmin-SNARE-membrane complex where the membrane interaction involves the polybasic region.

F) Brewer KD, et al. (2015) Nat Struct Mol Biol 22(7):555-564 which revealed additional possible interfaces and dynamic character of these additional interfaces that may be important upon calcium triggering.

G) Wang S, Li Y, Ma C (2016) *eLife* 5:1689-1699 which provided experimental evidence that Syt1 C2B indeed simultaneously interacts with PS/PIP2 containing membranes (via the polybasic region) and SNARE complex.

[Editors' note: further revisions were requested prior to acceptance, as described below.]

Thank you for resubmitting your work entitled "PtdInsP_2_ and PtdSer cooperate to trap synaptotagmin-1 to the plasma membrane in the presence of calcium" for further consideration at *eLife*. Your article has been favorably evaluated by Gary Westbrook (Senior Editor) and four reviewers, one of whom, Axel T Brunger, is a member of our Board of Reviewing Editors. The following individuals involved in review of your submission have agreed to reveal their identity: Timothy A Ryan (Reviewer #1); Jose Rizo-Rey (Reviewer #2).

We thank the authors for addressing many of the concerns raised in the previous review, but there are a few remaining issues. We kindly ask the authors to address these few remaining points that hopefully will be easily addressable.

1) In Figure 2, it is unclear how the polybasic region can interact with PIP2. It also seems that the orientation of C2B is different to that shown in Figure 7; if this is the case, the authors should clarify why they changed the orientation.

2) The authors argue that the binding of Syt1 to Pip2 is strong because, while the measured Kd is in the μM range, the local concentration of lipids seen by Syt1 in vivo will be much higher (subsection “PtdInsP_2_ binding to the C2B domain polybasic patch”, last paragraph). We agree that the high local concentration of lipids will favor binding, but the authors should make the same consideration with respect to SNARE complex binding to Syt1. Note that this issue is ignored in the aforementioned paragraph and again in the second paragraph of the Discussion where they mention the lack of SNARE complex binding to Syt1 at physiological ionic strength. The ITC data of Figure 4 indicate tighter binding of Syt1 to the SNARE complex than to InsP_3_; hence, it seems difficult to draw a strong conclusion that PIP2 binding should dominate in vivo, as the local concentrations of both PIP2 and SNARE complex will be much higher than in these experiments and both interactions can cooperate with PS binding to other parts of Syt1. We suggest that the authors make a more balanced discussion of the data and leave the issue of whether PIP2 or the SNARE complex 'wins' open.

3) In this revised version, the authors added ITC experiments involving C2B, C2AB and SNARE complex (Figure 4). Upon mutation of the two lysine residues in the polybasic region (KAKA mutant), the interaction apparently becomes weaker. However, as the authors point out, the ITC data are rather noisy, so they should be interpreted with caution. In fact, the group of one of the reviewers (ATB) tried to reproduce the result with the KAKA mutant, and found conditions that revealed a much more significant interaction between the mutated C2B domain and SNARE complex than Figure 4 might suggest, consistent with the primary interface observed by Zhou et al. (2015). However, since different constructs were used, this discrepancy requires further investigation that is probably outside the scope of this work.

4) At the end of the second paragraph of the Discussion it seems to imply that the importance of the Syt1 residues involved in the primary interface observed by Zhou et al., has not been tested. In fact, mutation of these residues disrupts fast synchronous release (Zhou et al). The authors are kindly asked to clarify their statement in the aforementioned paragraph.

---

## [Author Response]

[…]

For the non-specialist reader, it is difficult to determine the advances that have been made in this study because previous publications came to related conclusions, especially upon casual reading. Thus, the manuscript would benefit greatly by describing previous results (see for a partial list below) in a balanced fashion and then placing the results of this study into historic perspective. The reviewers also raised a number of substantive issues outlined below, and additional experiments are suggested to strengthen studies of the polybasic region, and to reconcile their results with another recently published study (Wang et al. eLife 2016).

Major comments:

1) The authors conclude that PIP2 binds to a specific binding site of the C2B domain. However, their NMR data (Figure 3) is in apparent contradiction with this conclusion. It is unlikely that the observed chemical shift changes (which span residues more than 30 angstroms apart) can be explained by PIP2 binding to a single site. The NMR data are much more easily explained by a model whereby PIP2 can quickly switch between different binding modes along the polybasic region (which includes six basic residues along a single β-strand plus additional basic residues in nearby strands), and also the calcium-binding region. This notion does not contradict the approximately 1:1 stoichiometry measured by the authors, as one single molecule of PIP2 could move around the polybasic region and its strong negative potential would hinder binding of a second PIP2 molecule to the region. The overall lack of a specific, defined binding site is supported by the lack of specificity for different stereoisomers of phosphatidylinositols shown by the authors (subsection “Phosphoinosites and Ca^2+^ decrease dissociation rate of synaptotagmin-1”, first paragraph).

We agree with the reviewer, the simplest interpretation of the NMR data is that the PIP2 headgroup may not have a single well-defined binding site on C2B. Although chemical shift mapping in NMR is not unequivocal, the data to not preclude the possibility that the PIP2 headgroup exchanges between multiple binding modes that are similar in energy. This is consistent with the electrostatic nature of the interaction and the fact that there is a lack of specificity for different isomers of this charged lipid headgroup. To make this point clear, we modified the statements within the Results and the Discussion so as not to mislead the reader.

2) The authors only mutate two residues (K326 and K327) among the many that form the polybasic region. Without comparing the effects of mutations in all the residues of the polybasic region, it is difficult to draw conclusions about specificity for a single, well-defined binding site. Studies with additional mutations would strengthen this part of the manuscript. And one should also consider that the side chains of K326 and K327 are on opposite sides of a β strand.

As pointed out above in 1), we agree that PIP2 may not interact at a specific well-defined site. Our goal was not to completely define the InsP_3_ binding region, but rather to demonstrate the role of the polybasic region of C2B domain in PIP_2_ binding. To this end, we decided to use the KAKA mutant because it has been widely studied and is reported to have a dramatic effect on PIP_2_ binding (Li et al., 2006, van den Bogaart et al., 2012). Our ITC results support this conclusion, showing that InsP_3_ binds specifically to the polybasic region determined by our NMR data. We do not think studying additional mutants in the polybasic region will add to the present study. The reviewer points out that sites K326 and K327 are on opposite sides of a β strand; however, since the strand is on the edge of a β-sandwich both residues are available to interact with the headgroup of PIP2 given the domain orientations shown in Figure 2.

3) In the second paragraph of the Discussion, the authors state that the calcium binding sites specifically bind PS but not phosphoinositides. However, it has been known for a long time that other negatively charged phospholipids such as π can bind similarly to the synaptotagmin-1 C2 domains; hence, the specificity for PS is quite relative. Note also that the NMR data (Figure 3) show that there is some degree of binding of PIP2 to the calcium-binding region, even though it is less populated than binding to the polybasic region.

We agree with the reviewer and we have changed the wording in the manuscript (Discussion, third paragraph) to indicate that the C2 domains exhibit a “preferential” rather than”specific” binding to PS. As explained in the Discussion (third paragraph), the use of IP3 as an analog to PIP2 isolates the PIP2 binding site in the NMR experiment without interference by a closely apposed membrane interface. Since IP3 is highly negatively charged, it is not surprising that there is a weak interaction of IP3 with the positively charged Ca^2+^-binding region in the absence of this membrane interface.

4) In the fourth paragraph of the subsection “PtdInsP_2_ binding to the C2B domain polybasic patch”, the authors describe ITC experiments showing that the presence of the SNARE complex does not alter binding to the soluble InsP_3_ (Figure 4), and conclude that this result supports the notion that synaptotagmin-1 binds to PIP2 preferentially over the SNARE complex, in contrast to conclusions drawn in Brewer et al., 2015, and also in contrast to the data presented in a recent paper (Wang S, Li Y, Ma C (2016) eLife 5:1689-1699). However, the conclusion from the authors was obtained with the soluble head group whereas Brewer et al., 2015 used liposomes containing 1% PIP2. In Radhakrishnan et al., 2009 it has been shown that binding of synaptotagmin-1 to liposomes containing 1% PIP2 is very weak (Radhakrishnan et al., 2009), and in this current paper the experiments performed with liposomes containing 2.5% PIP2 (no PS) also seem to suggest a very low affinity (subsection “PtdSer and PtdIns(4,5)P_2_ play a synergistic role in membrane binding of synaptotagmin-1”, second paragraph and Figure 1). Hence, these experiments with liposomes correlate better with the conclusions of ref. 25 than with the ITC results. It is also important to emphasize that the SNARE complex aggregates at the concentrations used in the ITC experiments (50 μM), which could influence the results obtained. The authors should discuss these issues and consider whether ITC experiments performed with InsP_3_ in the presence of the SNARE complex are conclusive, also in view of the recent paper by Wang, Li, and Ma (2016). Additional experiments may be needed to reconcile these different conclusions.

We would like to point out the data in Figure 5 and Table 2 actually suggest a very high affinity for PIP2 directed by the polybasic face, and we have included some discussion on page 10 to clarify this point. Furthermore, the results of a recent study from our labs indicates that PIP2-Syt1 interactions mediated by the polybasic face take precedence over SNARE interactions (Park, 2015). At this point, we cannot rule out that SNARE interactions occur at other sites on Syt1, and therefore our conclusions are not inconsistent with the recent paper by Wang (2016). Nonetheless, our results suggest that it is unlikely that Syt-1 interacts with the SNAREs through the polybasic face in the presence of PIP2. We have amended the Discussion to address these issues.

As suggested by the reviewer, we have carried out additional experiments using ITC to examine the interactions of IP3 with C2B and C2AB. We lowered the protein concentrations to 15 μm and do find that there is some reduction of IP3 affinity in the presence of the SNAREs. However, this appears to be associated with the presence of C2A (rather than the lowered concentration). We added the results of these experiments to Figure 4 and we addressed this result and the use of IP3 on page 14 in the Discussion.

5) The interpretation of the results in terms of static pictures is only a model. It is possible, or even likely that the interactions of the synaptotagmin-1 C2 domains with membranes are dynamic, and such an interpretation is perfectly compatible with the data presented by the authors in this paper.

Sorry for the misunderstanding – we completely agree that the lipid interface is clearly dynamic, and that the C2B domain will obviously sample a range of orientations. Since the EPR depth measurements yield averaged values, the models presented in Figure 7 are averaged structures. On the other hand, the domains are not likely to be undergoing large-scale changes in orientation. Such large scale motions would expose the calcium binding loops to both aqueous and hydrocarbon environments and this would yield distinct modes in the EPR line shapes that are not seen. So that the readers are not confused, we added a few comments to the last paragraph of the subsection “PtdSer and PtdIns(4,5)P_2_ play a synergistic role in membrane binding of synaptotagmin-1”, to address this point.

6) As mentioned above, the work lacks discussion of previous work, including (this is not meant to be a comprehensive list):

A) Bai J, Tucker WC, Chapman ER (2004) Nat Struct Mol Biol 11:36-44 which shows that the polybasic region of the Syt1 C2B domain interacts with PIP2 containing membranes without and with calcium.

B) Bai J, Wang P, Chapman ER (2002) Proc Natl Acad Sci U S A 99:1665-1670 which finds that the calcium binding loops of the Syt1 C2B domain insert into the membrane upon calcium binding.

C) Rufener E, Frazier AA, Wieser CM, et al. (2005) Biochemistry 44:18-28 which determined the orientation and position of the Syt1 C2B domain upon calcium binding into the membrane using EPR.

D) Van Bogaart G Den, Meyenberg K, Diederichsen U, Jahn R (2012) J Biol Chem 287:16447-16453 which shows that the polybasic patch of tagmin is critical in the protein's ability to bind PIP2.

E) Zhou Q, Lai Y, Bacaj T, et al. (2015) Nature 525:62-67 which revealed interfaces between synaptotagmin-1 and SNARE complex, one of which was found to be very important for function in the neuron, and suggested a model of a pre-assembled synaptotagmin-SNARE-membrane complex where the membrane interaction involves the polybasic region.

F) Brewer KD, et al. (2015) Nat Struct Mol Biol 22(7):555-564 which revealed additional possible interfaces and dynamic character of these additional interfaces that may be important upon calcium triggering.

G) Wang S, Li Y, Ma C (2016) eLife 5:1689-1699 which provided experimental evidence that Syt1 C2B indeed simultaneously interacts with PS/PIP2 containing membranes (via the polybasic region) and SNARE complex.

We revised the Results and Discussion sections to include references to some of the previous work as suggested by the reviewer. We particularly focused on those references related to SNARE-syt binding.

[Editors' note: further revisions were requested prior to acceptance, as described below.]

[…]

We thank the authors for addressing many of the concerns raised in the previous review, but there are a few remaining issues. We kindly ask the authors to address these few remaining points that hopefully will be easily addressable.

1) In Figure 2, it is unclear how the polybasic region can interact with PIP2. It also seems that the orientation of C2B is different to that shown in Figure 7; if this is the case, the authors should clarify why they changed the orientation.

We revised Figure 2 and Figure 7 to better show the C2B domain orientation while taking into account our EPR data as suggested by the reviewer.

2) The authors argue that the binding of Syt1 to Pip2 is strong because, while the measured Kd is in the μM range, the local concentration of lipids seen by Syt1 in vivo will be much higher (subsection “PtdInsP_2_ binding to the C2B domain polybasic patch”, last paragraph). We agree that the high local concentration of lipids will favor binding, but the authors should make the same consideration with respect to SNARE complex binding to Syt1. Note that this issue is ignored in the aforementioned paragraph and again in the second paragraph of the Discussion where they mention the lack of SNARE complex binding to Syt1 at physiological ionic strength. The ITC data of Figure 4 indicate tighter binding of Syt1 to the SNARE complex than to InsP_3_; hence, it seems difficult to draw a strong conclusion that PIP2 binding should dominate in vivo, as the local concentrations of both PIP2 and SNARE complex will be much higher than in these experiments and both interactions can cooperate with PS binding to other parts of Syt1. We suggest that the authors make a more balanced discussion of the data and leave the issue of whether PIP2 or the SNARE complex 'wins' open.

We have revised the Discussion in order to take these arguments into account. Clearly, there are different views on whether SNARE-binding or PIP_2_-binding is physiologically more relevant. During revision, we have spelled this out more clearly, but we believe that we are entitled to have our own opinion (obviously while respecting that of our colleagues thinking otherwise). This is a true scientific debate that we consider as fruitful. Yes, a Discussion needs to be balanced, and we hope to have achieved this better during revision, but we are entitled to our own views as long as they are justified by scientific reasoning.

3) In this revised version, the authors added ITC experiments involving C2B, C2AB and SNARE complex (Figure 4). Upon mutation of the two lysine residues in the polybasic region (KAKA mutant), the interaction apparently becomes weaker. However, as the authors point out, the ITC data are rather noisy, so they should be interpreted with caution. In fact, the group of one of the reviewers (ATB) tried to reproduce the result with the KAKA mutant, and found conditions that revealed a much more significant interaction between the mutated C2B domain and SNARE complex than Figure 4 might suggest, consistent with the primary interface observed by Zhou et al. (2015). However, since different constructs were used, this discrepancy requires further investigation that is probably outside the scope of this work.

We agree with the reviewer, and we would like to point out that the KAKA mutant didn’t abolish the syt-SNARE binding, but the low concentration of the proteins hindered the reliable determination of the binding affinity.

4) At the end of the second paragraph of the Discussion it seems to imply that the importance of the Syt1 residues involved in the primary interface observed by Zhou et al., has not been tested. In fact, mutation of these residues disrupts fast synchronous release (Zhou et al). The authors are kindly asked to clarify their statement in the aforementioned paragraph.

We agree with the reviewer, and we have amended it in the new Discussion.